# Diagnosis and prognosis prediction of gastric cancer by high-performance serum lipidome fingerprints

Ze-Rong Cai[1,6], Wen Wang [ID][2,3,6], Di Chen[2,6], Hao-Jie Chen [ID][1,6], Yan Hu [ID][1,6], Xiao-Jing Luo[1], Yi-Ting Wang[4], Yi-Qian Pan[1], Hai-Yu Mo[1], Shu-Yu Luo[1], Kun Liao[1], Zhao-Lei Zeng [ID][1], Shan-Shan Li[5], Xin-Yuan Guan[5], Xin-Juan Fan[4], Hai-long Piao [ID][2✉], Rui-Hua Xu [ID][1,5✉] & Huai-Qiang Ju [ID][1,5✉]

## Abstract

**Early detection is warranted to improve prognosis of gastric cancer (GC) but remains challenging. Liquid biopsy combined with machine learning will provide new insights into diagnostic strategies of GC. Lipid metabolism reprogramming plays a crucial role in the initiation and development of tumors. Here, we integrated the lipidomics data of three cohorts ($n = 944$) to develop the lipid metabolic landscape of GC. We further constructed the serum lipid metabolic signature (SLMS) by machine learning, which showed great performance in distinguishing GC patients from healthy donors. Notably, the SLMS also held high efficacy in the diagnosis of early-stage GC. Besides, by performing unsupervised consensus clustering analysis on the lipid metabolic matrix of patients with GC, we generated the gastric cancer prognostic subtypes (GCPSs) with significantly different overall survival. Furthermore, the lipid metabolic disturbance in GC tissues was demonstrated by multi-omics analysis, which showed partially consistent with that in GC serums. Collectively, this study revealed an innovative strategy of liquid biopsy for the diagnosis of GC on the basis of the serum lipid metabolic fingerprints.**

**Keywords** Biomarker; Diagnosis; Gastric Cancer; Lipid Metabolism; Prognosis
**Subject Categories** Biomarkers; Cancer

## Introduction

Gastric cancer is the fifth most common malignant tumor and the fourth-leading cause of cancer-associated mortality worldwide (Sung et al, 2021). The 5-year survival rate can exceed 90% for early-stage GC patients undergoing radical surgery, compared with advanced GC patients with chemotherapy and immunotherapy whose median survival was less than 15 months (Janjigian et al, 2021). Thus, the early detection and diagnosis of GC is of paramount significance for improving the prognosis of patients. However, traditional techniques such as serum-based biomarkers, radiological technology and tissue biopsies based on endoscopy have proven to be challenging (Elmore et al, 2021). For instance, previously reported biomarkers lack superior sensitivities and specificities while microscopic lesions are often missed with imaging results (Elsherif et al, 2020; Liu et al, 2020). Furthermore, as the current gold standard, gastroscopy combined with histo-pathological findings is expensive, invasive, risky and relies on the experience of endoscopists, which makes it unsuitable for the large-scale screening (Thrift and El-Serag, 2020). Overall, these phenomena emphasize the necessity of developing promising tools for the early detection of GC.

Liquid biopsy is an alternative tool for the minimally invasive detection of circulating tumor cells, circulating tumor DNA (ctDNA) and extracellular vesicles by body fluids (Bradley and Barclay, 2021). Previous studies, including ours, have indicated that liquid biopsy offers more opportunities for the diagnosis and prognosis monitoring of cancers (Guo et al, 2023; Ju et al, 2019; Luo et al, 2020). Importantly, liquid biopsy allows repeated sampling and is less influenced by cell heterogeneity compared to assessing specimens of tumor samples, which means highly valuable for liquid biopsy-based biomarker discovery. Despite several traditional blood-based biomarkers, including carcinoembryonic antigen (CEA), carbohydrate antigen 19-9 (CA19-9), and carbohydrate antigen 72-4 (CA72-4), have been widely applied in suggesting the presence of gastrointestinal tumors (Xu et al, 2023), they were inappropriate for the early screening of GC owing to their low sensitivities (Sekiguchi and Matsuda, 2020). Therefore, novel biomarkers from body fluids have sprung up, including ctDNA, extracellular vesicle-derived lncRNA GClnc1, and a serum 12-microRNA assay (Guo et al, 2023; Maron et al, 2019; So et al, 2021).

Aggressive tumor growth is characterized by significant metabolic reprogramming that provides opportunities for cancer diagnosis and therapeutics (Li et al, 2023a). Since lipids are the

[1]State Key Laboratory of Oncology in South China, Guangdong Provincial Clinical Research Center for Cancer, Sun Yat-sen University Cancer Center, Guangzhou 510060, P. R. China. [2]CAS Key Laboratory of Separation Science for Analytical Chemistry, Dalian Institute of Chemical Physics, Chinese Academy of Sciences, Dalian 116023, P. R. China. [3]Department of Neurology, The First Affiliated Hospital of Anhui Medical University, Hefei 230032, P. R. China. [4]The Sixth Affiliated Hospital, Sun Yat-sen University, Guangzhou 510655, P. R. China. [5]Department of Clinical Oncology, Shenzhen Key Laboratory for Cancer Metastasis and Personalized Therapy, The University of Hong Kong-Shenzhen Hospital, Shenzhen, P. R. China. [6]These authors contributed equally: Ze-Rong Cai, Wen Wang, Di Chen, Hao-Jie Chen, Yan Hu.✉E-mail: hpiao@dicp.ac.cn; xurh@sysucc.org.cn; juhq@sysucc.org.cn

main components of biofilms, energy donors, and signal transducers of cells (Keckesova et al, 2017), lipid metabolism plays a crucial role in the initiation and development of tumors (Minami et al, 2023). Furthermore, compared with nucleic acids and proteins, lipid metabolites can directly reflect the phenotype of cancers and provide real-time feedback of the human body as downstream molecules (Butler et al, 2020). Importantly, the development of lipidomics technology and the advances in machine learning have allowed for the discovery of cancer-specific signatures and the construction of statistical classifiers for screening in population (Capper et al, 2018; Chen et al, 2023b; Mayerle et al, 2018). Therefore, lipid metabolites are expected to become promising biomarkers and some liquid biopsy-based lipids have been reported in diagnosis of lung cancer and pancreatic cancer (Wang et al, 2022; Wolrab et al, 2022). Due to the importance of lipid metabolism and the gap in exploiting serum lipid signatures for the screening of GC, elucidating the role of lipid metabolites in the detection and prognosis evaluation of GC is extremely urgent.

Here, serum lipidomics data of GC patients and healthy donors from multiple cohorts were analyzed to portray the lipid metabolic landscape of GC. Novel tools of liquid biopsy for diagnosis and prognosis prediction of GC were developed through machine learning using lipid signatures and further affirmed in an external validation cohort and a predictive cohort. Notably, the SLMS outperformed the traditional biomarkers of gastrointestinal tumors in diagnosing GC patients, especially patients with early-stage GC. In addition, the GCPS can predict the survival of GC patients and subtype-specific genes may be beneficial for the individualized treatment of GC. Moreover, we also conducted multi-omics analyses on GC tissues, which confirmed the lipid metabolism disorder in GC and the rationality of the SLMS and GCPS. Collectively, our exploration refreshed our understanding of the lipid metabolic fingerprints of GC and the machine learning predictors were conducive to the early detection and precision medicine in GC.

# Results

## The depiction of GC lipid metabolic landscape

First, to investigate the metabolic reprogramming of GC, lipids and hydrophilic metabolites were detected in 28 GC patients and 28 healthy donors. As shown in Fig. EV1A, principal component analysis indicated that the difference of lipids was more significant than that of hydrophilic metabolites. Based on this interesting finding and the significance of lipid metabolism in cancers, we would like to describe the lipid metabolic landscape of GC.

As shown in the flow diagram in Fig. 1A, we first collected serums of 266 GC patients and 266 healthy donors from Sun Yat-sen University Cancer Center (SYSUCC) as an exploration cohort, of which 85% were used as the training cohort and 15% as the testing cohort. Subsequently, serum lipids were detected by ultra performance liquid chromatography-tandem mass spectrometry (UPLC/MS) (Xuan et al, 2020a; Xuan et al, 2020b). Quality control samples were prepared and identically inserted in the analytical sequence, of which the analysis could be seen in Fig. EV1B–D. To globally understand the lipid metabolic landscape of GC patients and healthy donors, the partial least squares-discriminant analysis

(PLS-DA) was performed on the metabolism data in the training cohort (Figs. 1B and EV1E) and the lipids were ranked by the variable importance projection (VIP) scores. The results showed the different lipid metabolic patterns between gastric cancer and health. This phenomenon was consistent with preliminary findings and previous research in other solid tumors (Huang et al, 2022; Wang et al, 2021).

Among the 581 detected lipids, the levels of 207 lipids were remarkably different between GC patients and healthy donors, including 36 phosphatidylcholine, 24 lyso-phosphatidylethanolamine, 22 ether-linked phosphatidylcholine, 18 sphingomyelin, 17 fatty acid and 90 others (Fig. EV1F). Pathway enrichment analysis revealed that alpha linolenic acid and linoleic acid metabolism, biosynthesis of unsaturated fatty acids, beta oxidation of very long chain fatty acids and glycerophospholipid metabolism changed along with the occurrence of GC (Fig. EV1G).

## Construction of the serum lipid metabolic signature for GC diagnosis

Machine learning has been applied to detect cancers by establishment of appropriate and effective markers (Capper et al, 2018). First, to check redundancy of lipid features, we performed a spearman's correlation analysis between detected lipids and revealed a high correlation between detected lipids, especially the top 50 lipids with the highest VIP scores (Fig. 1C,D). Therefore, we only included lipids with spearman's correlation coefficients less than 0.5 in later analysis. Subsequently, we utilized machine learning models to establish a diagnostic predictor for GC in the training cohort (Fig. 1E). Specifically, ten commonly used classification algorithms, including LDA, SVM Linear, SVM Linear Weights, SVM Radial, SVM Radial Weights, RF, KNN, Glmnet, Bayesglm, and QDA, were included in the selection (Li et al, 2023b) at the same time as the signature screening on the filtered top 50 lipids sequentially. Through 10-fold crossover validation, the predictor achieved the performance with a mean accuracy of 0.963 when the LDA algorithm (Li et al, 2022) was applied on the filtered top 19 metabolites (Fig. 1F), which made up the serum lipid metabolic signature (Fig. 1G, Table EV1). In general, we constructed a diagnostic predictor with the combination of SLMS and LDA algorithm for GC detection.

To verify whether this predictor is specific for classifying GC patients and healthy donors, we calculated the scores of LDA-aided SLMS from the probability of each sample being diagnosed as GC. As shown in Fig. EV2A,B, the scores of GC patients showed no significant difference across different stratifications of various clinical characteristics in the training cohort. In addition, univariate and multivariate linear regression analyses were carried out on all GC cases or healthy cases in the training cohort. None of the clinical characteristics achieved significant *P* values in the multivariate model (Table EV2). Finally, the scores were significantly higher in GC patients (average score of 0.939) than in healthy donors (average score of 0.044, Fig. EV2C) and only the SLMS score was an independent predictive factor of GC status in the multivariate model (Table EV3). These results shed new light that the SLMS was independent of clinical characteristics when used for diagnosing GC and highlighted the potential clinical transformation of the SLMS.

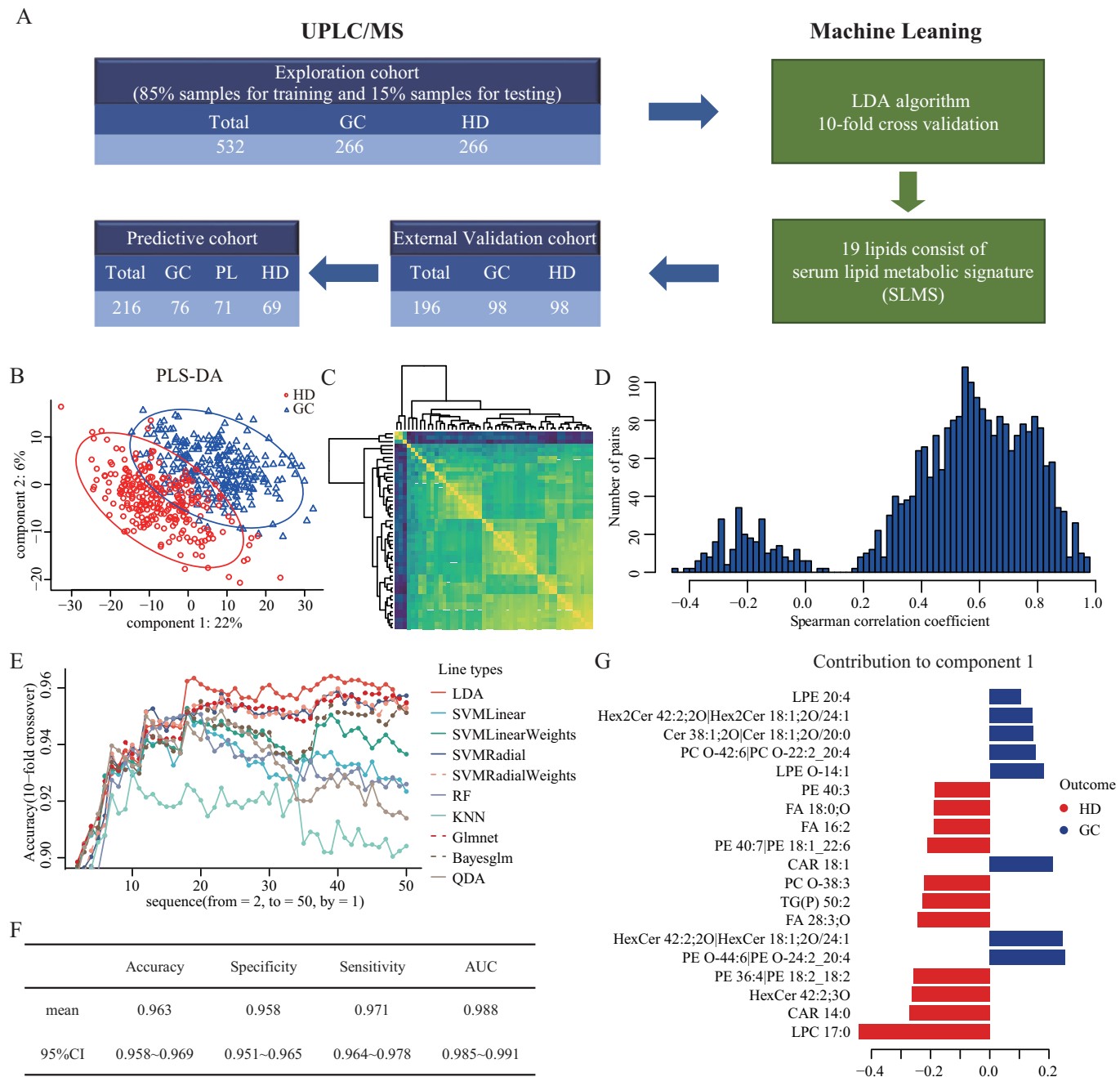

**Figure 1. Construction of the serum lipid metabolic signature for GC diagnosis.**

(A) Flow diagram for the construction and validation of SLMS. (B) Partial least squares-discriminant analysis of serum lipidomics between GC patients and healthy donors in the training cohort. (C) Heatmap for correlation analysis of the expression of the top 50 lipid metabolites. (D) Histogram of the distribution of spearman's correlation coefficients between metabolites. (E) Accuracies of 10 classification algorithms when using different numbers of metabolites. (F) Results of 10-fold crossover validation of lipidomics data from the training cohort by using LDA and top 19 lipids. (G) The 19 lipids of SLMS and their contribution to component 1, ranking small to large. Bayesglm bayesian generalized linear models, GC gastric cancer, Glmnet lasso and elastic-net regularized generalized linear model, HD healthy donor, KNN k-nearest neighbor, LDA linear discrimination analysis, PL precancerous lesion, PLS-DA partial least squares-discriminant analysis, QDA quadratic discriminant analysis, RF random forest, SLMS serum lipid metabolic signature, SVMLinear linear support vector machine, SVMLinearWeights linear support vector machine with class weights, SVMRadial support vector machine with radial basis function, SVMRadialWeights support vector machine with radial basis function and class weights, UPLC/MS ultra-high performance liquid chromatography/mass spectrometry. Source data are available online for this figure.

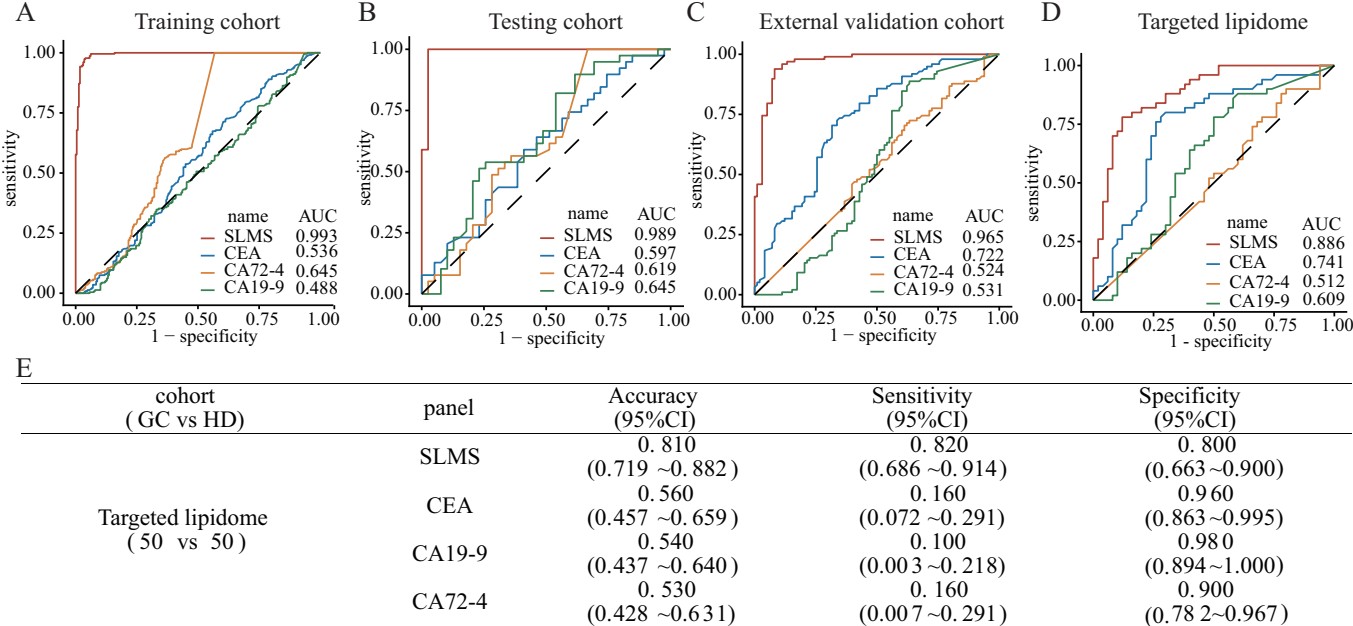

**Figure 2.  Validation of the diagnostic efficacy of SLMS.**

(A–C) The ROC curves of SLMS, CEA, CA19-9, and CA72-4 to differentiate gastric cancer patients (all stages) from healthy donors in the training, testing and external validation cohorts. (D, E) The diagnostic performance of SLMS in targeted lipidomics data. AUC area under curve, CA19-9 carbohydrate antigen 199, CA72-4 carbohydrate antigen 724, CEA carcinoembryonic antigen, CI confidence interval, HD healthy donor, ROC receiver operating characteristic, SLMS serum lipid metabolic signature.

Although the scores of SLMS were independent of pTNM stages (Fig. EV2A), the 19 lipids in SLMS showed 3 significantly distinct trends (Cluster 1–3) along with the progression of GC pathological stage (Fig. EV2D). Specifically, the lipids in Cluster 1 (e.g., HexCer 42:2;2O|HexCer 18:1;2O/24:1) exhibited a continuously increasing pattern while those lipids in Cluster 2 (e.g., LPC 17:0) showed a sustainably decreasing trend along with cancer development. Besides, some lipids, such as PE O-44:6|PE O-24:2_20:4, partially declined after climbing up during the progression of GC. This finding was beneficial to the specific mechanism study of the advancement in gastric cancer.

## Validation of the diagnostic efficacy of SLMS

To assess the efficacy of the LDA-aided SLMS, the receiver operating characteristic (ROC) curve was employed. The result revealed that the LDA-aided SLMS achieved a desirable area under curve (AUC) of 0.993, an accuracy of 0.967, a sensitivity of 0.960 and a specificity of 0.974 in the training cohort (Fig. 2A, Table 1). To our knowledge, abnormalities of some peripheral blood tumor markers, including CEA, CA19-9 and CA72-4, may indicate the presence of gastric cancer (Xu et al, 2023). Therefore, we also examined the performance of these markers in our cohorts. The results showed that the efficacy of SLMS was superior to that of CEA (AUC, 0.536; accuracy, 0.520; sensitivity, 0.075; specificity, 0.965), CA19-9 (AUC, 0.488; accuracy, 0.518; sensitivity, 0.154; specificity, 0.881), CA72-4 (AUC, 0.645; accuracy, 0.518; sensitivity, 0.154; specificity, 0.881) and the combination of these three biomarkers (accuracy, 0.579; sensitivity, 0.260; specificity, 0.837; Fig. 2A, Table 1). To evaluate the robustness and accuracy of the

SLMS, the testing cohort mentioned above (39 GC patients and 39 healthy donors) and an external validation cohort from the Gastrointestinal and Anal Hospital of Guangdong Province (98 GC patients and 98 healthy donors) were subjected to serum lipidomics. The results exhibited that the SLMS generated an AUC value of 0.989 with an accuracy of 0.974, a sensitivity of 0.949 and a specificity of 1.000 in the testing cohort, and an AUC value of 0.965 with an accuracy of 0.913, a sensitivity of 0.929 and a specificity of 0.898 in the external validation cohort (Fig. 2B,C; Table 1). The 10-fold crossover validation and successful validation in dependent cohorts indicated that there was no overfitting in SLMS. Meanwhile, CEA, CA19-9, CA72-4 and the combination of these three biomarkers still revealed low AUC values, accuracies, sensitivities, and specificities in the other two cohorts (Fig. 2B,C; Table 1). Furthermore, the SLMS was verified to be an independent predictor of GC in the testing and external validation cohorts (Fig. EV2A–C).

Notably, to further validate our LDA-aided SLMS, we constructed a targeted lipidomics method for 19 lipids in SLMS, and then detected the expression of each lipid in serums from 50 GC patients and 50 healthy donors. Through the calculation of LDA algorithm, SLMS achieved good diagnostic performance with an AUC of 0.886, an accuracy of 0.810, a sensitivity of 0.820 and a specificity of 0.800 (Fig. 2D,E). These verification efforts confirmed the earlier findings and the clinical application value of the SLMS.

Previous study has reported that 20–40% of GC patients are negative for CEA, CA19-9, and CA72-4 on account of the Lewis a⁻b⁻ genotype, hence resulting in missed diagnosis (Guo et al, 2023). To investigate the diagnostic efficacy of the SLMS for this subgroup, we screened out GC patients with negative CEA, CA19-9, and CA72-4 from each cohort, and assessed the performance of SLMS when used

**Table 1.** Classification performance of SLMS and other traditional gastrointestinal tumor-related biomarkers in GC patients of different cohorts.

| Cohort | Panel | Accuracy (95%CI) | Sensitivity (95%CI) | Specificity (95%CI) |
|---|---|---|---|---|
| Training cohort | SLMS | 0.967 (0.946–0.981) | 0.960 (0.926–0.982) | 0.974 (0.943–0.990) |
| | CEA | 0.520 (0.473–0.567) | 0.075 (0.044–0.117) | 0.965 (0.932–0.985) |
| | CA19-9 | 0.540 (0.493–0.586) | 0.093 (0.058–0.138) | 0.987 (0.962–0.997) |
| | CA72-4 | 0.518 (0.471–0.564) | 0.154 (0.110–0.208) | 0.881 (0.832–0.920) |
| | CDP | 0.549 (0.501–0.595) | 0.260 (0.204–0.322) | 0.837 (0.782–0.883) |
| Testing cohort | SLMS | 0.974 (0.910–0.997) | 0.949 (0.829–0.994) | 1.000 (0.910–1.000) |
| | CEA | 0.551 (0.434–0.664) | 0.128 (0.043–0.274) | 0.974 (0.865–0.999) |
| | CA19-9 | 0.538 (0.422–0.652) | 0.103 (0.029–0.242) | 0.974 (0.965–0.999) |
| | CA72-4 | 0.500 (0.985–0.615) | 0.154 (0.059–0.305) | 0.846 (0.695–0.941) |
| | CDP | 0.564 (0.447–0.676) | 0.308 (0.170–0.476) | 0.821 (0.665–0.925) |
| External validation cohort | SLMS | 0.913 (0.865–0.949) | 0.929 (0.859–0.971) | 0.898 (0.820–0.950) |
| | CEA | 0.561 (0.489–0.632) | 0.143 (0.080–0.228) | 0.980 (0.928–0.998) |
| | CA19-9 | 0.566 (0.494–0.637) | 0.143 (0.080–0.228) | 0.990 (0.945–1.000) |
| | CA72-4 | 0.515 (0.443–0.587) | 0.153 (0.088–0.240) | 0.878 (0.796–0.935) |
| | CDP | 0.622 (0.551–0.691) | 0.398 (0.300–0.502) | 0.847 (0.760–0.912) |

The combined diagnostic panel (CDP) was proposed based on CEA, CA19-9 and CA72-4. Specifically, participants who were positive for any of these three biomarkers would be identified as patients with gastric cancer.

*CA19-9* carbohydrate antigen 199, *CA72-4* carbohydrate antigen 724, *CEA* carcinoembryonic antigen, *CI* confidence interval, *GC* gastric cancer, *SLMS* serum lipid metabolic signature.

for judging them. As shown in Fig. EV2E,F, the SLMS revealed excellent diagnostic performance with AUCs of 0.993, 0.985, and 0.965 in the training, testing, and external validation cohorts, respectively. Overall, the result indicated that the SLMS was robust in identifying GC patients with negative CEA, CA19-9, and CA72-4.

## Application of the SLMS in the early detection of GC

Early detection is definitely warranted to improve prognosis in GC patients (Ma et al, 2023). Gastric cancer, especially the intestinal type, develops from a cascade of precancerous lesions in a stepwise progression, so patients with precancerous lesions are at high risk of GC (Correa, 1992). Identifying cases of precancerous lesions is significant for executing efficient prevention and management strategies. Therefore, we further collected serum samples from 71 patients with gastric precancerous lesions from SYSUCC to constitute a predictive cohort in the validation phase. First, we examined the abundance of 19 lipids in SLMS in three groups of healthy donors, patients with precancerous lesions and GC patients. The results delineated that the abundance of some lipids increased (Fig. 3A) or decreased (Fig. 3B) in a stepwise manner from normal to precancerous lesions and then to GC, which suggested that these lipids may serve as evident biomarkers of disease progression. In addition, we investigated the performance of SLMS in the predictive cohort. Consistent with previous results, the SLMS still achieved high diagnostic efficacy with an AUC of 0.977 in differentiating GC patients ($n = 76$) from healthy donors ($n = 69$, Fig. 3C). However, the SLMS only had an AUC of 0.782 with a sensitivity of 0.606 in distinguishing patients with gastric precancerous lesions from healthy donors (Fig. 3C, Table EV4). Consequently, the SLMS may tend to identify precancerous lesions as GC and was beneficial to filter out the patients who need further gastroscopy examination.

It is well known that an efficient biomarker is capable of distinguishing patients with early-stage gastric cancer (EGC, pTNM stage I/II), so we further evaluated the diagnostic performance of the SLMS in patients with EGC. Primarily, the scores of LDA-aided SLMS in EGC patients (average of 0.928, 0.903, 0.946, and 0.863, respectively) were remarkably higher than those of healthy donors (average of 0.044, 0.029, 0.133, and 0.086, respectively) in the training, testing, external validation and predictive cohorts (Fig. EV2G). In addition, the SLMS had the best AUCs (0.992, 0.996, 0.975, and 0.963, respectively) to discriminate EGC patients from healthy donors in comparison with CEA, CA19-9 and CA72-4s in all four cohorts (Fig. 3D–G). Subsequently, by using the cut-off value of 0.5, the accuracy for identifying EGC patients was 0.964, the sensitivity was 0.945 and the specificity was 0.974 in the training cohort (Table 2). The performance of the SLMS was then verified in the other three cohorts, where accuracies for identifying EGC patients were 0.984, 0.922, and 0.923, respectively; sensitivities were 0.958, 0.977, and 0.886, respectively; and specificities were 1.000, 0.898, and 0.942, respectively (Table 2). Moreover, the SLMS showed similar high-performance during different pTNM stages (Table EV5). In conclusion, these data elucidated that the SLMS could identify patients with EGC successfully and held great application potential for early screening in population.

## Serum lipid metabolites can predict the prognosis of GC

As previous studies reported that serum markers can predict the prognosis of tumors (Lee et al, 2023; Ma et al, 2023), we further explored the application of serum lipids in the prognostic prediction of GC. As shown in Fig. 4A, the GCPS indicated by serum lipid metabolites was constructed by using the lipid metabolic data matrix of GC patients in the exploration cohort

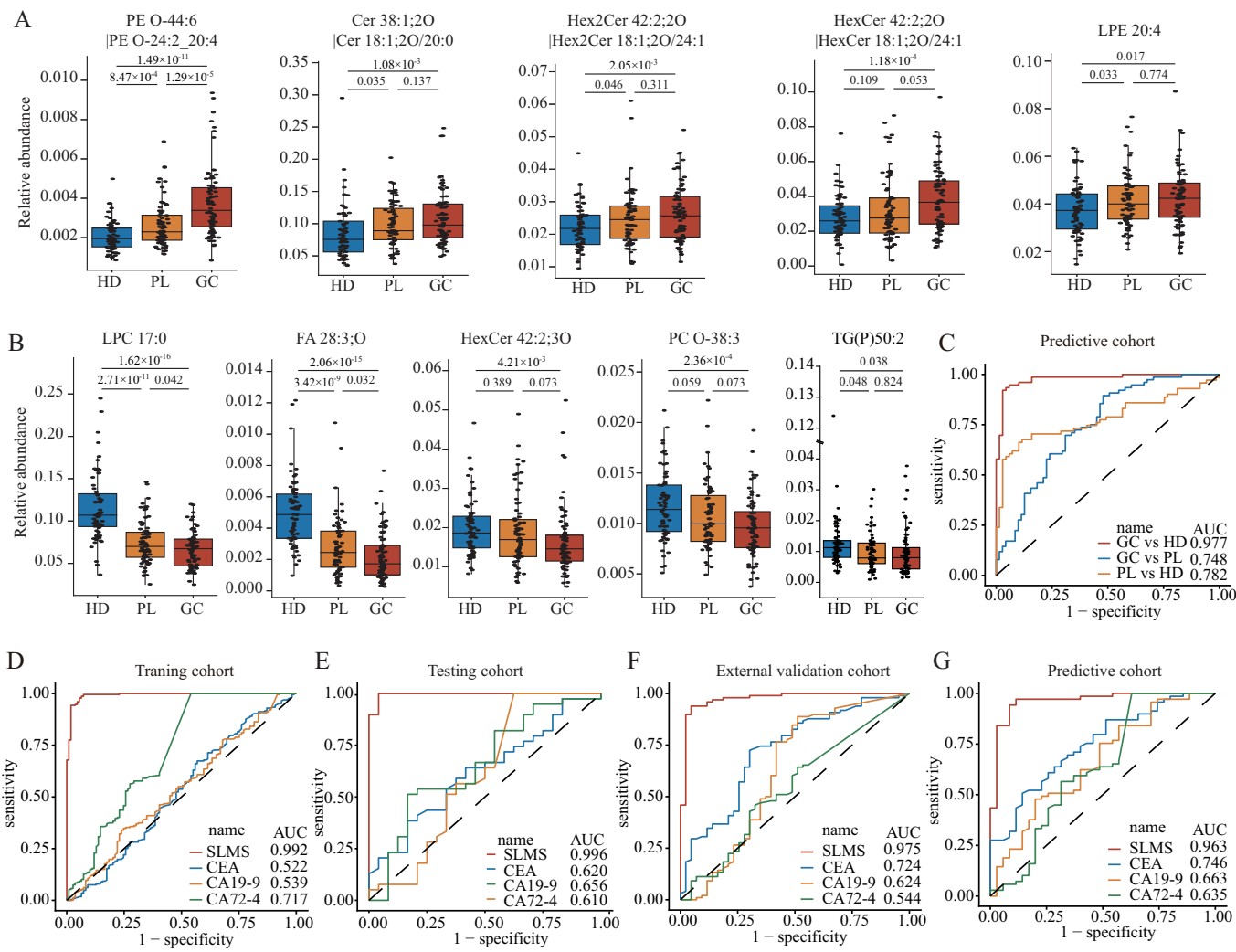

**Figure 3. Application of the SLMS in the early detection of GC.**

(A, B) The significantly downregulated (A) and upregulated (B) metabolites during the process from HD turning to PL and GC finally (Student's *t* test). The numbers of the participants in HD, PL, and GC groups were 69, 71, and 76, respectively. (C) The ROC curves of SLMS in comparing any two groups in the predictive cohort. (D–G) The ROC curves of SLMS, CEA, CA19-9, and CA72-4 to differentiate GC patients in early-stage from healthy donors in the training (D), testing (E), external validation (F), and predictive cohorts (G). In the box plots of A and B, the upper bound, the line inside and the lower bound shows the 75th, 50th, and 25th percentiles of the sample while whiskers are extended to the most extreme data point that is no more than 1.5× interquartile range (75th percentile minus 25th percentile) from the edge of the box. AUC area under curve, CA19-9 carbohydrate antigen 199, CA72-4 carbohydrate antigen 724, CEA carcinoembryonic antigen, GC gastric cancer, HD healthy donor, PL precancerous lesion, ROC receiver operating characteristic, SLMS serum lipid metabolic signature.

(266 observations with 54 deaths and a median follow-up time of 33 months). Subsequently, we confirmed the generalization ability of the GCPS in the external validation cohort (98 observations with 30 deaths and a median follow-up time of 46 months) and the predictive cohort (76 observations with 17 deaths and a median follow-up time of 32.5 months). Specifically, we identified 2 metabolic subtypes (SI-SII) by using a consensus clustering method on the metabolic profile of lipids associated with prognosis in the exploration cohort (Fig. 4B). By generating Kaplan–Meier curves, we found SI was with significantly worse prognosis than SII with two-year overall survival (OS) rates of 75.7% and 88.8% for SI and SII, respectively (hazard ratio = 3.34, 95% CI :1.912–5.848, *P* < 0.001, log-rank test; Fig. 4C, Table EV6). Afterward, a subtype

predictor was trained by the Glmnet method (Zuccato et al, 2023) and employed to predict the subtypes of the samples in the external validation and predictive cohorts. Then, clinical analysis and mechanism analysis were conducted to assess the differences between subtypes. It turned out that SI was related to a larger maximum diameter (*P* = 0.001, *P* = 0.012 and *P* < 0.001, respectively; Chi-square test; Fig. EV3A, Table EV7) and advanced pTNM stage (*P* < 0.001, *P* = 0.022 and *P* = 0.003, respectively; Chi-square test; Fig. EV3B, Table EV8) in the exploration, external validation, and predictive cohorts. In addition, there were significant differences in the expression of lipids between the two subtypes (Fig. EV3C), which were mainly enriched in the pathways of biosynthesis of unsaturated fatty acids, glycerophospholipid

**Table 2.** Classification performance of SLMS and other traditional gastrointestinal tumor-related biomarkers in EGC patients of different cohorts.

| cohort (EGC vs HD) | panel | Accuracy (95%CI) | Sensitivity (95%CI) | Specificity (95%CI) |
|---|---|---|---|---|
| Training cohort (109 vs 227) | SLMS | 0.964 (0.938-0.981) | 0.945 (0.884-0.980) | 0.974 (0.943-0.990) |
| | CEA | 0.673 (0.620-0.723) | 0.064 (0.026-0.128 | 0.965 (0.932-0.985 |
| | CA19-9 | 0.676 (0.623-0.725) | 0.028 (0.006-0.008) | 0.987 (0.962-0.997) |
| | CA72-4 | 0.619 (0.565-0.671) | 0.077 (0.003-0.140) | 0.881 (0.832-0.920) |
| Testing cohort (24 vs 39) | SLMS | 0.984 (0.915-1.000) | 0.958 (0.789-0.999) | 1.000 (0.910-1.000) |
| | CEA | 0.651 (0.520-0.767) | 0.125 (0.027-0.324) | 0.974 (0.865-0.999) |
| | CA19-9 | 0.619 (0.488-0.739) | 0.042 (0.001-0.211) | 0.974 (0.865-0.999) |
| | CA72-4 | 0.603 (0.472-0.724) | 0.201 (0.071-0.422) | 0.846 (0.695-0.941) |
| External validation cohort (43 vs 98) | SLMS | 0.922 (0.865-0.960) | 0.977 (0.877-0.994) | 0.898 (0.820-0.50) |
| | CEA | 0.702 (0.619-0.776) | 0.070 (0.001-0.190) | 0.980 (0.928-0.998) |
| | CA19-9 | 0.716 (0.634-0.789) | 0.093 (0.026-0.221) | 0.990 (0.945-1.000) |
| | CA72-4 | 0.660 (0.575-0.737) | 0.163 (0.068-0.307) | 0.878 (0.796-0.935) |
| Predictive cohort (35 vs 69) | SLMS | 0.923 (0.854-0.966) | 0.886 (0.732-0.968) | 0.942 (0.858-0.984) |
| | CEA | 0.702 (0.604-0.788) | 0.143 (0.005-0.303) | 0.986 (0.922-1.000) |
| | CA19-9 | 0.692 (0.594-0.779) | 0.086 (0.018-0.231) | 1.000 (0.948-1.000) |
| | CA72-4 | 0.654 (0.554-0.745) | 0.171 (0.066-0.337) | 0.899 (0.802-0.958) |

*CA19-9* carbohydrate antigen 199, *CA72-4* carbohydrate antigen 724, *CEA* carcinoembryonic antigen, *CI* confidence interval, *EGC* early-stage gastric cancer, *HD* healthy donor, *SLMS* serum lipid metabolic signature.

metabolism and linoleic acid metabolism (Fig. EV3D). Generally, we are the first to develop a prognostic subtyping method for GC based on serum lipid metabolites.

Subsequently, we continued to evaluate the prognostic prediction efficacy of the GCPS. As shown in Fig. 4C and Table EV6, GC patients with SI had worse OS than those with SII in the external validation cohort and the predictive cohort, which suggested that we did not overfit the prognostic prediction model. Notably, multivariable Cox regression analysis with the GCPS and clinicopathological characteristics confirmed that the GCPS was a consistently independent predictive factor of OS (exploration cohort: hazard ratio = 2.224, 95% CI: 1.141–4.336, $P = 0.019$; external validation cohort: hazard ratio = 3.566, 95% CI: 1.355–9.388, $P = 0.010$; predictive cohort: hazard ratio = 6.386, 95% CI: 1.676–24.336, $P = 0.007$; Fig. EV3E, Table EV8). Furthermore, the GCPS could also stratify the survival of patients with advanced GC (AGC, pTNM III/IV). The OS for AGC patients with SI was strikingly shorter than that for those with SII ($P = 0.014$, $P = 0.003$, and $P = 0.002$ for the exploration, external validation, and predictive cohorts, respectively; log-rank test; Fig. 4D). In summary, we validated the GCPS as a valid predictive system for OS in GC patients and extended its application in the prognostic prediction of GC.

## Multi-omics reveals a global lipid metabolism disturbance in GC

The transformations of metabolism in patients' blood on account of metabolic reprogramming of cancer cells have been extensively reviewed in previous studies, disclosing that metabolomic information in the blood may partly reflect the existence of tumors and prognosis of cancer patients (Su et al, 2021; Wang et al, 2021). To comprehend the lipidome profiling of GC tissues, we conducted lipidome and spatial metabolome analyses on GC tissues and normal tissues. First, according to the lipidome results, eight lipids of SLMS displayed significant differences in abundance between GC tissues and normal tissues while all lipids of SLMS were significantly dysregulated in the serum of GC patients in comparison with healthy donors (Fig. EV4A, Table EV9). As a supplementary, the expression of five lipids from SLMS in the serums of GC patients was partially or completely restored to normal owing to surgical resection of tumor tissues (Fig. EV4B, Table EV10). Moreover, the spatial metabolome results revealed that there were different metabolic patterns among the tumor, paratumor, and normal regions (Figs. 5A,B and EV5). Changes of lipid metabolites, including glycerophospholipids, glycerolipids, fatty acids, and sphingolipids were also observed between the tumor region and the normal region (Figs. 5C,D and EV4C). Importantly, three lipids in the SLMS could be detected by the spatial metabolome, and their expression was changed in GC tissues in situ (Fig. 5E). Finally, transcriptomics analysis (The Cancer Genome Atlas Research Network, 2014; Data ref: The Cancer Genome Atlas Research Network, 2014) and proteomics analysis (Shi et al, 2023; Data ref: Shi et al, 2023) of GC tissues were conducted to systematically explore lipid metabolic reprogramming in GC. The results suggested that quite a few metabolic pathways related to lipids were dysregulated in GC (Figs. 5F and EV4D). Notably, there were changes in gene expression and protein levels related to the pathways enriched by the different lipids in serums between GC patients and healthy donors, which were shown in Fig. EV1F and Table EV11. Based on analysis of multi-omics data, we elucidated the profile of disturbed lipid metabolism in GC tissues and confirmed the rationality of SLMS in predicting GC.

Since multi-omics analysis indicates the reprogramming of lipid metabolism in GC tissues, we speculated that the metabolic

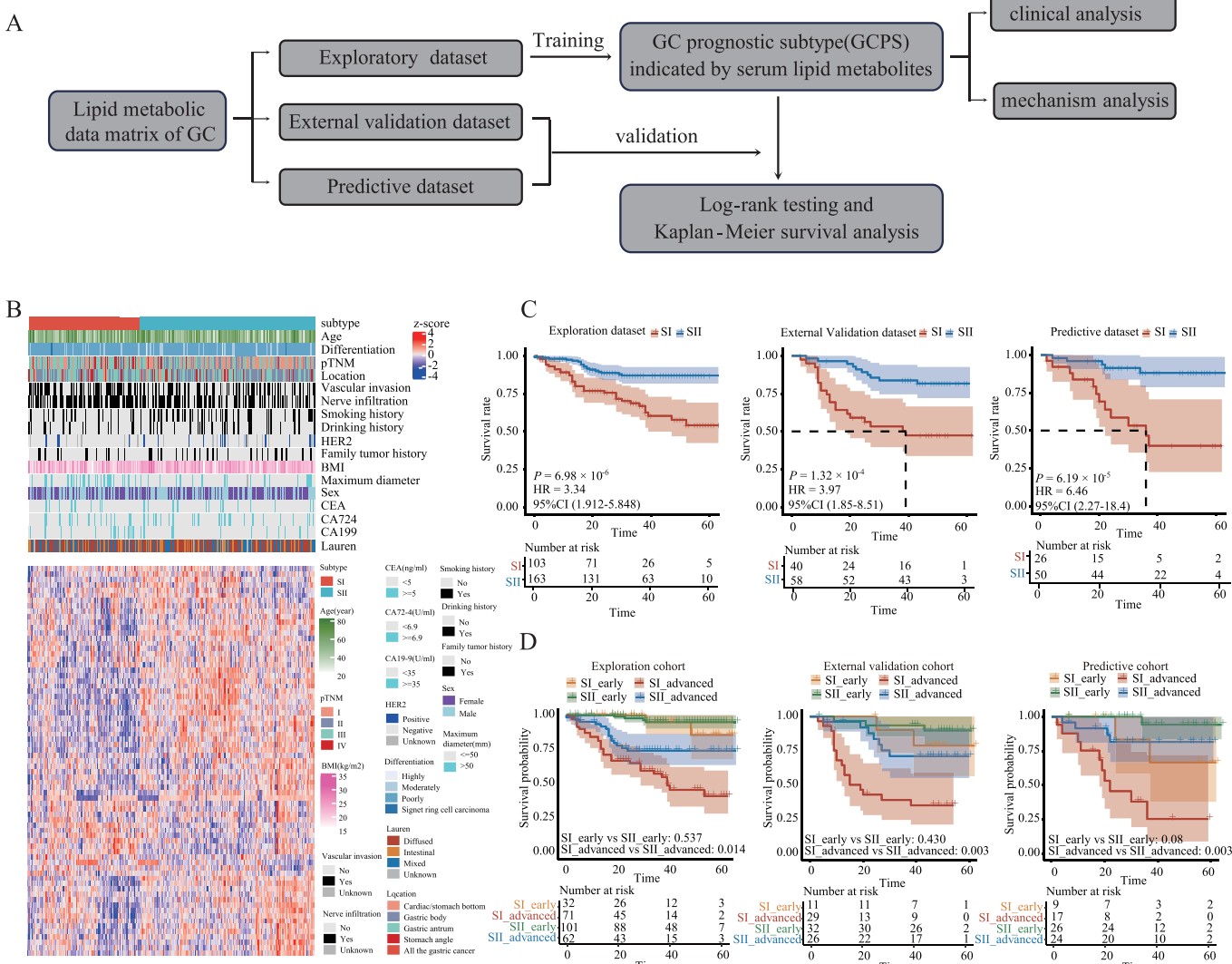

**Figure 4. Serum lipid metabolites can predict the prognosis of GC.**

(A) Workflow for the building and validation of GCPS. (B) Patient subgrouping based on the lipids associated with prognosis (*P* < 0.05 for univariable variant cox analysis and spearman's correlation coefficients less than 0.5 between lipids). Samples and lipid metabolites are displayed as columns and rows, respectively, and the color of each cell shows the *z*-score of the relative abundance of the lipids (logarithmic scale in base 2). (C) Kaplan–Meier curves for OS based on the GCPS for the exploration, external validation, and predictive cohorts. (D) Prognostic analysis of the GCPS in GC patients with different stages in the exploration, external validation, and predictive cohorts. *P* values were determined by log-rank test (C, D). CI confidence interval, GC gastric cancer, GCPS gastric cancer prognostic subtype, HR Hazard ratio, OS overall survival.

features were also different between GC tissues from SI and SII. Thus, we further conducted transcriptome analysis on GC tissues from different subtypes. Pathway enrichment of differentially expressed genes revealed that metabolism-related pathways, including lipid metabolism, were significantly different between the two prognostic subtypes, which confirmed the conclusion of lipidomics (Fig. 5G, Table EV12). Eventually, to explore potential subtype-specific drug targets, we searched the DGIdb database for genes with significant differences between the two subtypes. Some genes associated with lipid metabolism and lipid transport were identified, such as *CHAT* and *FFAR1* in SI and *APOB*, *APOC3*, and *MTTP* in SII (Hooper et al, 2005; Zhang et al, 2022) (Fig. 5H, Table EV13). To sum up, we further verified the metabolic

differences in two prognostic subtypes and identified some gene targets that were beneficial to subtype-specific therapy.

# Discussion

It has been reported that ctDNAs and exosome-derived non-coding RNAs could act as novel diagnostic biomarkers for GC (Guo et al, 2023; Maron et al, 2019). Nevertheless, a small amount of ctDNAs in circulation require sensitive and expensive detection techniques while the fragmented exosome-derived ncRNAs could not reflect the overall genetic profile of the tumors. In contrast to DNA and RNA, lipids could directly reflect the holistic and real-time

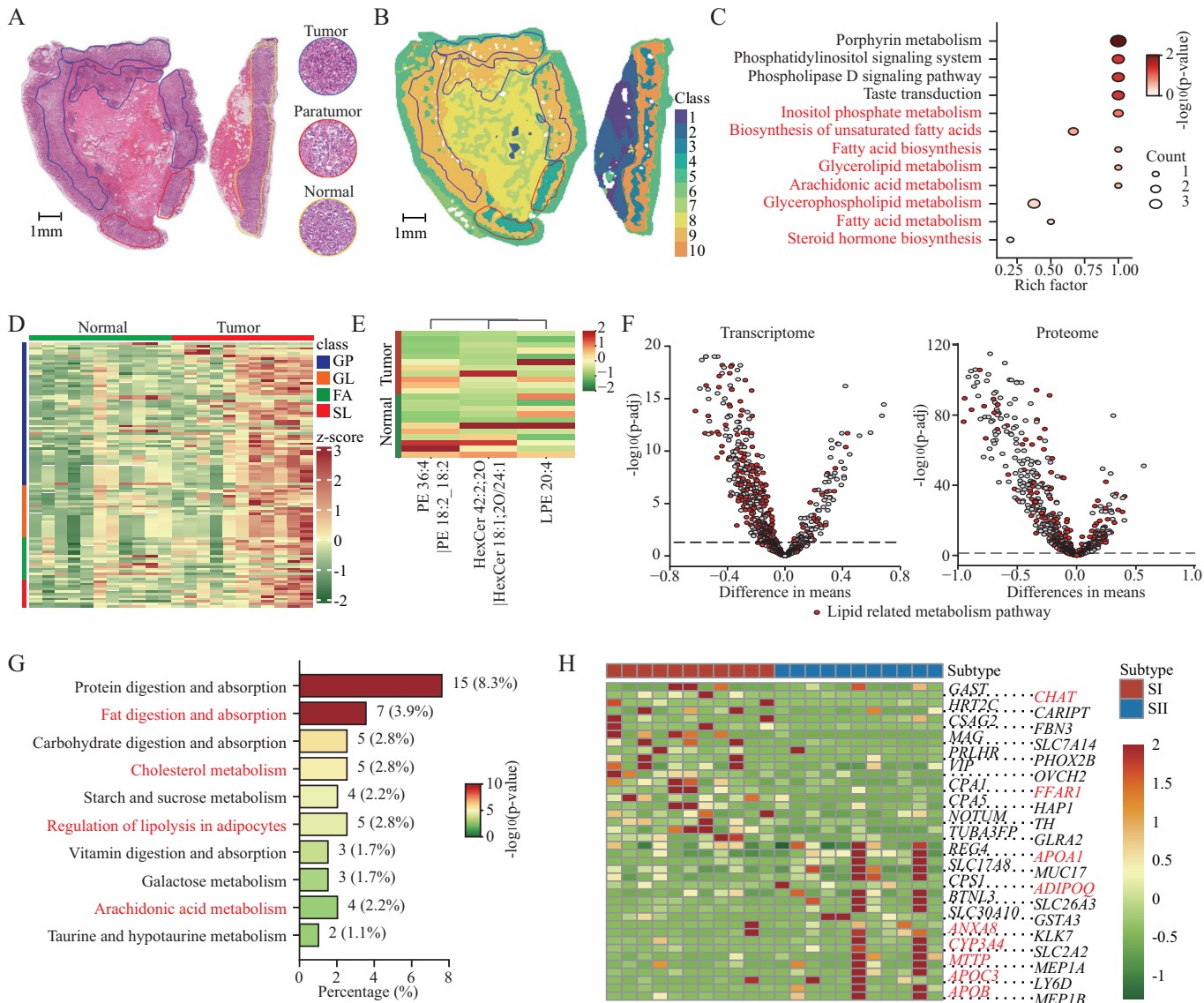

**Figure 5. Multi-omics analysis reveals a global lipid metabolism disturbance in GC.**

(A) The H&E stain image of tissue section and magnified images of different regions (×20), scale bar = 1 mm. The region of gastric cancer, paratumor, and normal were encircled by blue, red, and yellow lines, respectively. (B) The metabolite-driven segmentation of tissue section based on the metabolome data, scale bar = 1 mm. (C) Enrichment analysis of differentially expressed metabolites between the GC regions and the normal regions. (D) Heatmap showing the lipids that were differentially expressed between GC and normal regions. (E) A hierarchical clustering plot and heatmap showing the global abundance of lipids in SLMS between GC and normal tissues. (F) Volcano plots showing the changes of metabolic pathways in GC through transcriptome analysis (380 GC tissues vs 37 normal tissues) and proteome analysis (194 GC tissues vs 194 normal tissues). (G) Bar plot of the KEGG pathway map enriched by differentially expressed genes between two prognostic subtypes. (H) Heatmap of differentially expressed druggable targets between prognostic subtypes. The pathways (G) and genes (H) in red color are associated with the metabolism and transport of lipids. The expression levels of metabolites or genes in (D), (E), and (H) are represented as row-normalized z-scores. P values were determined by Hypergeometric test (C, G), Student's t-test (D, F), Paired t-test (E), and Deseq2 (H). FA fatty acids, GC gastric cancer, GP glycerophospholipids, GL glycerolipids, H&E hematoxylin and eosin, KEGG Kyoto Encyclopedia of Genes and Genomes, p-adj adjusted P value for multiple testing via the Benjamini-Hochberg procedure, SL sphingolipid, SLMS serum lipid metabolic signature.

phenotypes of the tumors. Besides, the detection method of lipids, such as UPLC/MS, was universal and low-cost. Recent studies have shown that lipidomics could be utilized in the diagnosis of various cancers (Wang et al, 2022; Wang et al, 2021; Wolrab et al, 2022). In parallel, the SLMS and GCPS we established in this study exhibited great performances in the diagnosis and prognostic prediction of GC. Clinically effective management of cancers demands the

integration of early detection with risk stratified intervention. The SLMS could be used in recognizing potential GC patients, who would be confirmed by endoscopic biopsy finally. This strategy could improve the accuracy and efficiency of early screening. Subsequently, GC patients were classified into two groups with high and low risk of poor survival through GCPS. The postoperative therapeutic regimen of them will be formulated with overall

consideration of pTNM stages and GCPS. In general, the innovative lipid fingerprints were valuable and promising for the diagnosis and prognosis of GC.

Accumulating evidence has demonstrated the crucial role of precancerous lesions in the stepwise progression model of the initiation of gastric cancer and has provided some markers during this evolution (Sugano et al, 2023). A recent study has identified plasma metabolomic signatures in precancerous gastric lesions that progress to GC (Huang et al, 2021) and an extracellular vesicle-derived lncRNA GClnc1 has been proven to accurately differentiate between GC and gastric precancerous lesions (Guo et al, 2023). Moreover, the lipid panel consisting of PC 44:5, PC 35:6e, and SM d40:3 was verified to be valuable for colorectal advanced adenoma to colorectal cancer sequence (Chen et al, 2023a). In this study, it was observed that several lipids in the SLMS presented a gradually decreasing or increasing tendency during the progression from normal to gastric precancerous lesions and then to GC. This result implied that lipid metabolism reprogramming is accompanied by alterations in the abundance of lipids in blood when normal tissues start to become malignant and emphasized the necessity of concentrating on these gradually altering lipids. Therefore, a multicenter prospective study with sufficient precancerous lesion patients can be conducted to explore the diagnostic capacity of the lipid signatures. In summary, we preliminarily explored the expression of lipids in patients with gastric precancerous lesions and expanded the possibility of using lipids to recognize precancerous lesions.

The pTNM staging system, based on the findings of imaging and pathology, has remained most crucial to treatment guidance and prognostication for GC (Fang et al, 2018). However, the clinical outcomes and prognosis of AGC patients are still very different even if they are at the same stage and accept similar treatment because of the higher tumor heterogeneity in AGC (Smyth et al, 2020). This finding reminded clinicians of reconsidering the current strategy in recommending postoperative adjuvant therapy according to pTNM stage. In addition, GC patients with different molecular characterizations were summarized into different The Cancer Genome Atlas (TCGA) classifications, which could suggest different prognoses and provide a roadmap for trials of targeted therapies [42]. From another perspective, we constructed the GCPS which could better stratify GC patients, especially AGC patients, into different prognostic subtypes. The GCPS not only increases the accuracy of survival prediction before surgery but also provides guidance for postoperative therapy. Under the same stage, patients with SI need more treatment and closer follow-up. Furthermore, we analyzed the potential druggable targets that can be beneficial for subtype-specific therapy. In conclusion, we demonstrated that the lipid metabolic fingerprint can be used for prognosis stratification and may assist in more appropriate clinical management.

In this study, we integrated the lipidome, spatial metabolome, transcriptome, and proteome to describe the landscape of lipid metabolism in GC tissues. Based on the systematic analysis, glycerophospholipid metabolism, linoleic acid metabolism and biosynthesis of unsaturated fatty acids were related to the occurrence and development of GC. Previous studies have shown that dysregulation of the glycerophospholipid metabolism drove cancer cell proliferation and mediates immune reflection

(Henderson et al, 2019; Saito et al, 2022). Linoleic acid was reported to prompt cancer cells to die and potentiate CD8 + T-cell metabolic fitness thus promoting antitumor immunity (Nava Lauson et al, 2023). Biosynthesis of unsaturated fatty acids was involved in maintaining lipid homeostasis during tumor progression (Terry et al, 2023). These findings implied that targeting lipid metabolism in tumors represents a highly promising therapeutic strategy for GC.

However, there are still some limitations in our study. First, more functional verification studies of lipids are needed to elucidate the underlying biological mechanism and provide novel intervention targets. Second, it should be noted that the levels of serum metabolites are affected by various confounding factors, including diet, gut microbiota and lifestyle (Bose et al, 2020). Besides, our study solely focuses on the East Asian population in South China. Therefore, a prospective cohort with a more rigorous design, including taking the aforementioned factors into account and recruiting more participants with diverse geographical locations and racial backgrounds, will be needed to transform our results into clinical practice. Generally, this first-time utility of lipid metabolic fingerprint for GC diagnosis and prognosis prediction will require further investigation.

In conclusion, we are the first to perform a comprehensive portray program of lipid metabolic landscape of GC, including serums and tissues. We also provide a kind of innovative lipid-based tool of liquid biopsy for the diagnosis and prognosis predicting of GC, which is pioneering in the field of the clinical application of gastric cancer lipid metabolism.

# Methods

**Reagents and tools table**

| Reagent/Resource | Reference or Source | Identifier or Catalog Number |
|---|---|---|
| **Experimental Models** | | |
| Patient samples | Sun Yat-sen University Cancer Center and Gastrointestinal and Anal Hospital of Guangdong Province | Tables EV2–3; Source data for Fig. 1 |
| **Recombinant DNA** | | |
| **Antibodies** | | |
| **Oligonucleotides and other sequence-based reagents** | | |
| **Chemicals, Enzymes and other reagents** | | |
| carnitine C16:0-d3 | Sigma-Aldrich | Cat#55107 |
| palmitic acid-d3 | Sigma-Aldrich | Cat#615951 |
| Cer d18:1-d7/18:0 | Avanti | Cat#860677P |
| LPC 17:0-d5 | Avanti | Cat#855679L |
| PC 17:0/22:4-d5 | Avanti | Cat#855678L |
| PE 17:0/17:0 | Avanti | Cat#830756P |
| SM d18:1/15:0-d9 | Avanti | Cat#860686P |
| TG 15:0/18:1/15:0-d5 | Avanti | Cat#860901P |

| Reagent/Resource | Reference or Source | Identifier or Catalog Number |
|---|---|---|
| **Software** | | |
| Mass Spectrometry-Data Independent Analysis (MS-DIAL) software (v4.9) | open access | |
| Analyst (v1.6) | AB SCIEX | |
| R software (v 4.0.5) | Open access | |
| mixOmics R package (6.14.1) | Rohart et al, 2017 | |
| caret R package (v 6.0-88) | Kuhn, 2021 | |
| pROC R package (Version 1.18.0) | Open access | |
| ConsensusClusterPlus R package (v 1.54.0) | Wilkerson and Hayes, 2010 | |
| ComplexHeatmap R package | Gu et al, 2016 | |
| survival R package (v 3.2-7) | Therneau, 2020 | |
| MetaboAnalyst (v.5.0) | Cardinal 2.14.0 | |
| **Other** | | |
| ACQUITY UPLC system | Waters | |
| tripleTOF™ 5600 plus mass spectrometer | AB SCIEX | |
| hybrid QQQ-linear ion trap mass spectrometer, Q-Trap 5500 system | AB SCIEX | |
| Q Exactive HF mass spectrometer | Thermo Fisher Scientific | |
| DGIdb database | Griffith et al, 2013 | |

## Ethics approval

All samples were collected with the informed consent from the donors and approval by the Human Genetic Resources Management Office of China (permit number, [2023] CJ0426). This study was conducted in line with the clinical protocol approved by the Institutional Research Ethics Committee of SYSUCC (Guangzhou, China, permit number B2022-769-01). This study conformed to the principles set out in the World Medical Association Declaration of Helsinki and the Department of Health and Human Services Belmont Report.

## Patient enrollment

In this study, a retrospective cohort that recruited a total of 944 participants was built from 2003 to 2021. We have used G*power 3.1.9.7 for a priori estimation of the sample sizes and our study has met the need of sample numbers for training models. All participants had their blood drawn after fasting for at least 8 h. Specifically, computer generated random numbers were used to assign 85% of the samples from the exploration cohort into the training cohort and 15% of the samples into the testing cohort. For GC patients, the inclusion criteria included (1) confirmed by pathological examination, (2) serum samples prior to surgery, (3) not receiving any neoadjuvant therapy, including medication, and

(4) complete clinical data and follow-up. For patients with gastric precancerous lesions, the inclusion criteria included: (1) pathological results consistent with the definition of gastric precancerous lesions from the guideline of the American Society for Gastrointestinal Endoscopy, (2) no cancer. Notably, some patients with precancerous lesions have more than one kind of precancerous lesion at the same time. The health statuses of healthy donors could be confirmed by the medical examination results and consulting medical history. For healthy donors, the inclusion criteria included: (1) no precancerous or malignant gastric lesions confirmed by endoscopy, (2) no tumor history or gastric precancerous lesion history. Baseline clinicopathological data, including age, sex, tumor maximum diameter, tumor location, tumor differentiation, pTNM stage, vascular invasion, nerve infiltration, human epidermal growth factor receptor 2 (HER2), smoking history, drinking history, family tumor history, body mass index, CEA, CA19-9, and CA72-4, were collected. The pathologic staging was based on the Union for International Cancer Control (UICC) Tumor-Node-Metastasis (TNM) staging system (8th edition). The definition of overall survival was the duration from the surgery time to death from any cause. Patients without known events were reviewed at the date of last confirmed follow-up. The clinical characteristics of all participants could be found in the Source Data for Fig. 1A and Tables EV2–3. Besides, the inclusion criteria of serum samples after surgery included (1) from the same patients as preoperative samples, (2) within three months after surgery, (3) before any postoperative adjuvant therapy.

## Serum sample preparation

Samples were collected according to standard procedures and stored in the cancer biobanks of SYSUCC and Gastrointestinal and Anal Hospital of Guangdong Province. Afterward, a chloroform/methanol/water system was applied to extract hydrophilic metabolites and lipids from the samples. Specifically, for lipidome analysis, 450 μL of methanol with internal standards (including carnitine C16:0-d3, palmitic acid-d3, Cer d18:1-d7/18:0, LPC 17:0-d5, PC 17:0/22:4-d5, PE 17:0/17:0, SM d18:1/15:0-d9, and TG 15:0/18:1/15:0-d5) was added to 50 μL of each serum sample. The mixture was vortexed followed by the addition of 700 μL chloroform. After phase breaking using 200 μL water and centrifugation (13,000 × g, 4 °C, 15 min), two aliquots of 320 μL hydrophobic layer were collected and freeze-dried for subsequent positive ion mode and negative ion mode detection, respectively. Notably, all the concentration of internal standards in extraction solvent for serum samples were 0.4 μg/mL. Quality control (QC) samples were prepared by using mixed serum samples from participants. For the detection of hydrophilic metabolites in serum, capillary electrophoresis-mass spectrometry (CE-MS) analysis was employed. Detailed CE-MS methods were performed as previously described (Zeng et al, 2014).

## Lipidomics

Untargeted lipidomics of serum samples was performed by an ACQUITY UPLC system (Waters) coupled with a tripleTOF™ 5600 plus mass spectrometer (AB SCIEX) as a previous study described (Xuan et al, 2020b). Briefly, lyophilized samples were reconstituted in chloroform/methanol (2:1, v/v) and diluted

threefold in ACN/IPA/H2O (65:30:5, v/v/v) containing 5 mM ammonium acetate. The C8 AQUITY column (2.1 mm × 100 mm × 1.7 μm) was used for lipid separation. The mobile phases consisted of 3:2 (v/v) ACN/H₂O (10 mM ammonium acetate, phase A) and 9:1 (v/v) IPA/ACN (10 mM ammonium acetate, phase B). The flow rate was set as 0.3 mL/min and the column temperature was 60 °C. The elution gradient started at 50% B, was held at this concentration for 1.5 min, was linearly increased to 85% B at 9 min, reached 100% B at 9.1 min and was held at this concentration for 1.9 min. Finally, the elution gradient was returned to 50% B within 0.1 min and held at this concentration for 1.9 min for equilibration. The total run time was 13 min. The ion spray voltage for mass spectrum (MS) was set at 5500 V and 4500 V in positive and negative ion modes, respectively. The interface heater temperature was 500 °C and 550 °C in positive and negative ion modes, respectively. Ion source gas 1, ion source gas 2, and curtain gas were set at 50, 50, and 35 psi in positive ion mode and 55, 55, and 35 psi in negative ion mode, respectively. The MS scan range was 150–1250 Da in both positive mode and negative mode. The MS/MS fragmentation patterns were acquired using an information-dependent analysis; the collision energy was set to 30 V (positive mode) and −30 V (negative mode) with a collision energy spread of 10 V. QC samples were identically inserted in the analytical sequence after every run of 20 serum samples to monitor the reproducibility of the analytical method. The analysis of quality control data could be found in Fig. EV1. Untargeted lipidomics was also employed to detect postoperative serum samples in Fig. EV4B.

## Data processing

According to instructions (Tsugawa et al, 2015), raw data were processed using mass spectrometry-data independent analysis (MS-DIAL) software (v4.9). The following parameters were set: (data collection) RT begin, 0 min; retention time end, 13 min; mass range begin, 150 Da; mass range end, 1250 Da; (peak detection) mass slice width, 0.1 Da; smoothing method, linear weighted moving average; smoothing level, 3 scan; minimum peak width, 5 scan; exclusion mass list, none; (alignment) retention time tolerance 0.1 min (positive mode) and 0.2 min (negative mode). Default values were used for other parameters. Lipid features were obtained with mass-to-charge ratio ($m/z$), retention time and MS/MS pattern by searching acquired MS/MS spectra against the internal MS/MS LipidBlast library in MS-DIAL program (Tsugawa et al, 2020). In this study, accurate mass tolerance (MS1) and accurate mass tolerance (MS2) were 0.01 Da and 0.025 Da, respectively. For lipids without ionic fragments but appeared in the full scan mass spectrogram, the identity of these lipid candidates was further confirmed by comparing the relative retention time between the known lipids and the candidate peaks within the same lipid class. For specific MS/MS fragments used to assign lipid class, for example, 184.0739 was selected as a characteristic product ion for PC, LPC, PC-O, LPC-O, and SM. 369.3516 was selected as a characteristic product ion for CE. 266.2791, 264.2635, 262.2479, and 312.326 were selected as characteristic product ions of Cer or HexCer with skeletons d18:0, d18:1, d18:2, and d20:0, respectively. 241.0119 was selected as a characteristic product ion for PI. For acyl chains, 251.2016, 283.2642, 281.2485, 279.2329, 303.2325, 301.2169, and 327.2329 were taken as the characteristics of

appearance of fatty acyls 16:2, 18:0, 18:1, 18:2, 20:4, 20:5, and 22:6, respectively. After peak integration and normalization, lipid features with the RSD < 30% were selected for subsequent analyses. Manual assignment was performed to examine detailed lipid structural information on characteristic ions, neutral loss of certain groups, and/or fatty acyl ionic fragments. For lipids with distinct characteristic ions, fatty acyl ionic fragments, and/or neutral loss of certain groups in the MS/MS spectrum, structural compositions of lipids were annotated (e.g., PC 38:6|PC 18:2_20:4). For those without fatty acyl fragments but with only information on characteristic ions and/or neutral loss of specific groups, they would be annotated with total no. of carbons and double bonds of acyl chains, e.g., PC 42:6. The relationship between retention time and number of acyl chain carbon or number of acyl chain double bond was further confirmed within the same lipid class.

## Feature selection

The relative expression of lipids was normalized by the corresponding internal standards and then transformed to the base-2 logarithm plus 5. Data generated by both positive-ion and negative-ion modes were merged together and scaled. PLS-DA were performed based on the scaled and pre-processed metabolism data of the training cohort and the lipids were ranked based on the VIP scores. The SCCs between the expressions of two lipids were calculated. To avoid duplicated information for training machine learning models, we proposed a step-wise feature selection strategy. Initially, the top-ranked lipid was selected. Then, the next lipid along the rank was checked sequentially and included only if the SCCs were less than 0.5 between this lipid and all the already included lipids; otherwise, this lipid was removed.

## Construction of the machine learning model

The classification model was constructed based on the expression of filtered top-ranked lipids (with the selected number of lipids tried from 2 to 50) in the training cohort in a 10-fold crossover validation manner (10 repeats) using the caret R package (v 6.0-88). Ten common classification models including k-nearest neighbor (KNN), random forest (RF), lasso and elastic-net regularized generalized linear model (Glmnet), linear discriminant analysis (LDA), quadratic discriminant analysis (QDA), Bayesian generalized linear model (Bayesglm), linear support vector machine (SVMLinear), SVM with Radial basis function (SVMRadial), support vector machine with radial basis function (SVMRadialWeights), and linear support vector machine with class weights (SVMLinearWeights) were compared, where parameters of each algorithm were optimized by crossover validation and grid-search strategies. Concretely, the hyper-parameters of each algorithm, if any, were optimized by the train function in the caret package with default settings, this function can fit predictive models over different tuning parameters. The best tuning parameters for all algorithms were listed in Source Data for Fig. 1E. The comparisons were according to the average accuracies of the 10-fold crossover validation analysis based on the training cohort. Finally, the LDA algorithm was selected and the score of the diagnostic predictor can be calculated as the following equation:

$$\text{score.lda}(j) = \sum_k \alpha_k \cdot metabolite_{k,j} + \alpha_0$$

where $\alpha_k$ represents the coefficient corresponding to the $k$th metabolite, and $metabolite_{k,j}$ represents the scaled and pre-processed expression value of the $k$th metabolite in patient $j$, $\alpha_0$ represents a constant. Samples with a score greater than 0.5 were judged as originating from GC patients; otherwise, they were judged as originating from healthy donors (Capper et al, 2018).

Based on the subtypes of samples from the exploration cohort, Glmnet algorithm was employed to train a subtype predictor and the equation of the Glmnet algorithm was shown as follows.

$$\min_{\beta_0, \beta} \frac{1}{N} w_i l(y_i, \beta_0 + \beta^T x_i) + \lambda \left[ \frac{(1-\alpha)||\beta||_2^2}{2} + \alpha ||\beta||_1 \right]$$

where $\beta$ represents the coefficient vector, $x_i$ represents the pre-processed metabolism profiles of the patient $i$, and $w_i$ is the observation weight (default = 1). $l(yi, \eta i)$ is the negative log-likelihood contribution for observation $i$. The *elastic net* penalty is controlled by $\alpha$ and bridges the gap between lasso regression ($\alpha = 1$, the default) and ridge regression ($\alpha = 0$). The tuning parameter $\lambda$ controls the overall strength of the penalty. The score of each subtype calculated by the subtype predictor signified the probability of the sample being confirmed as each subtype. The sample was finally identified as the subtype of which the score was higher.

The diagnostic predictor and the subtype predictor can be obtained from the link (https://github.com/diChen310/ML_GC) where investigators can get the instructions and methods for use simultaneously.

## Serum targeted lipidomics

An ACQUITY UPLC system (Waters) coupled with a hybrid QQQ-linear ion trap mass spectrometer, Q-Trap 5500 system (AB SCIEX) was used for UPLC/QQQ MRM MS-based targeted lipidomics analysis. The LC conditions for targeted lipidomics analysis were the same as those in the nontargeted lipidomics analysis mentioned above. For MS detection, the QQQ MRM MS was operated with IonSpray Voltage 5500 V in positive mode, the temperature was set as 500 °C; both Ion Source Gas 1 (GS1) and Ion Source Gas 2 (GS2) were set as 50. In negative mode, the QQQ MRM MS was operated with IonSpray Voltage −4500 V; the temperature was set as 550 °C; both GS1 and GS2 were set as 40. The Collision Gas and Curtain Gas were set as 'high' and 35, respectively, in both modes. The declustering potential (DP) and collision energy (CE) of lipid ion pairs were optimized as described before (Xuan et al, 2018) and then the precursor ion (Q1), characteristic product ion (Q3), lipid name (ID), optimized DP, and CE were imported into the MS acquisition method. The lipids monitored were processed with the Analyst software in Explore Mode and Quantitate Mode (version 1.6, AB SCIEX).

## Tissue lipidomics

GC tissues and control tissues were taken from 10 GC patients during surgery and then subjected to untargeted lipidomics. For tissue samples, sheared tissues were weighed and then 500 μL methanol with internal standards were added. Mixed grinding apparatus (Scientz-24) was used for homogenization (35 Hz, 1 min) followed by addition of 500 μL chloroform and vortex for 30 s. After phase breaking using 200 μL water and centrifugation

($13,000 \times g$, 4 °C, 15 min), 240 μL hydrophobic layer was collected and freeze-dried for lipidomics analysis. At the same time, the QC sample was prepared by combining the hydrophobic layer from each sample and then vacuum dried. The same chromatographic separation method was performed on a Waters ACQUITY UPLC system coupled with a Q Exactive HF mass spectrometer (Thermo Fisher Scientific). Specifically, the ion spray voltages were 3.5 and 3.0 kV in positive and negative ion modes, respectively. The full MS scan range was 120–1600 Da with a resolution setting of 120,000. The automatic gain control (AGC) target value was $3 \times 10^6$; the maximum injection time was 200 ms in full-scan MS and their values were $1 \times 10^5$ and 50 ms in MS/MS scans. The resolution setting for tandem mass spectra was 60,000. The stepped normalized collision energy (NCE) was set to 15%, 30%, and 45%. The capillary temperature and aux gas heater temperature were 300 °C and 350 °C, respectively. The aux gas and sheath gas were set as 10 psi and 45 psi, respectively. The S-lens RF level was 50. The raw data were normalized by corresponding internal standards and tissue weights.

## Spatial metabolome and transcriptome analysis

Tumor tissues and normal tissues (5 centimeters away from the margin of the tumor) were taken from 11 GC patients during surgeries and sent for matrix-assisted laser desorption/ionization–mass spectrometry imaging (Dufresne et al, 2019). Metabolite extraction, metabolite detection and data analysis were performed by Wuhan Metware Biotechnology Co., Ltd. (Wuhan, Hubei, China). Metabolite identification was performed by comparing the MS and MS/MS spectrum information with the in-house database and the Human Metabolome Database (HMDB). Besides, extracted adducted ions without MS/MS fragmentation information were imported into in-house database and the HMDB and annotated for lipid species according to molecular weight with an error < ±10 ppm. In addition, all confirmed target peaks were used to conduct spatially-aware nearest shrunken centroids clustering to obtain ten partition. As shown in Fig. 5B and Fig. EV5, each partition was represented by a color and different color-labeled partitions represented different metabolic patterns. Furthermore, adjacent tissue sections of samples undergoing mass spectrometry image (MSI) were stained with hematoxylin-eosin.

GC tissues from 10 SI patients and 10 SII patients were collected. Total RNA was extracted using TRIzol reagent (Invitrogen) after the tissues were broken with a homogenizer. RNA-seq and data analysis were conducted by Wuhan Metware Biotechnology Co., Ltd. (Wuhan, Hubei, China). Potential subtype-specific drug targets were investigated based on the genes with significant differences between the two subtypes ($P < 0.05$) and the DGIdb database (Griffith et al, 2013).

## Statistical analysis

Sample size was chosen based on the need for statistical power. Two-sided test was used in the statistical analysis and $P < 0.05$ was defined as significant. All analysis scripts were programmed using R software (v 4.0.5), with the mixOmics R package (6.14.1) for PLS-DA (Rohart et al, 2017), the caret R package (v 6.0-88) for machine learning (Kuhn, 2021), the pROC R package (Version 1.18.0)

**The paper explained**

**Problem**

Early detection is of great significance for improving prognosis of patients with gastric cancer (GC). Liquid biopsy is a revolutionary tool for early diagnosis and prognosis prediction of GC. In addition, lipids can directly reflect the cancer phenotype and provide real-time feedback on the human body's condition. However, the landscape of lipid metabolism in GC remains unknown and there are few studies on the application of lipid detection through liquid biopsy in the diagnosis and prognosis of GC.

**Results**

We constructed the serum lipid metabolic signature (SLMS) for GC diagnosis based on the lipidomics data from a large-scale cohort and the SLMS was demonstrated to exhibit excellent diagnostic performance in distinguishing GC patients from healthy donors with accuracies exceeding 0.9 in multiple cohorts. Then, we developed the gastric cancer prognostic subtypes (GCPSs), which could successfully classify GC patients into groups with good or poor prognosis. Last, multi-omics analysis revealed the global lipid metabolism disturbance of GC both in serums and tissues.

**Impact**

Our current study fills the gap in the study of serum lipid metabolism in GC and provides a promising liquid biopsy-based tool for the diagnosis and prognosis prediction of GC. The proposed SLMS and GCPS might assist in early screening and appropriate clinical management of GC.

for ROC curve analysis, the ConsensusClusterPlus R package (v 1.54.0) for consensus clustering method (Wilkerson and Hayes, 2010), the ComplexHeatmap R package for drawing of heatmap (Gu et al, 2016), the survival R package (v 3.2-7) for the log-rank test and univariant and multivariant Cox analysis (Therneau, 2020), the MetaboAnalyst (v.5.0) for pathway enrichment analysis and the R package (Cardinal 2.14.0) for analysis of metabolite spatial distribution and spatial segmentation. In addition, the specificity, sensitivity, and accuracy were calculated based on MedCalc (https://www.medcalc.org/calc/diagnostic_test.php) while pathway enrichment was performed on the website (https://www.metaboanalyst.ca/MetaboAnalyst/upload/EnrichUploadView.xhtml).

## Graphics

The images of "serum collection" and "UPLC/MS" in synopsis were created with BioRender.com.

## Data availability

The lipidomics data generated in this study could be available in the link (www.ebi.ac.uk/metabolights/MTBLS9126) (Yurekten et al, 2024), and this data could also been accessed in the OMIX (https://ngdc.cncb.ac.cn/omix) under accession No. OMIX007487. The spatial metabolomics would be made available on the link (https://metaspace2020.eu/project/cai-2023); The HE images of samples that underwent the spatial metabolomics are available in zenodo (https://doi.org/10.5281/zenodo.13744125). RNA-seq data

generated in this study are available in Genome Sequence Archive HRA005690 (http://bigd.big.ac.cn/gsa-human/).

The source data of this paper are collected in the following database record: biostudies:S-SCDT-10_1038-S44321-024-00169-0.

## Peer review information

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

## Acknowledgements

We appreciate the great technical support from the Sun Yat-sen University Cancer Center for their sample collection and follow-up of patients. We appreciate Wuhan Metware Biotechnology Co., Ltd. (Wuhan, Hubei, China) for their technical support. This project was supported by the National Key R&D Program of China (2022YFA1105300), National Natural Science Foundation of China (82341010, 82273241, 81972625, 82103643, 32201217), Guangdong Basic and Applied Basic Research Foundation (2023B1515040030), Liaoning Revitalization Talents Program (XLYC2002035), Sanming Project of Medicine in Shenzhen (SZSM202211017) and Young Talents Program of Sun Yat-sen University Cancer Center (YTP-SYSUCC-0019).

## Author contributions

**Ze-Rong Cai**: Data curation; Formal analysis; Writing—original draft. **Wen Wang**: Data curation; Formal analysis; Methodology; Writing—original draft. **Di Chen**: Formal analysis; Methodology; Writing—original draft. **Hao-Jie Chen**: Formal analysis; Writing—original draft. **Yan Hu**: Data curation; Writing—original draft. **Xiao-Jing Luo**: Formal analysis. **Yi-Ting Wang**: Data curation. **Yi-Qian Pan**: Investigation. **Hai-yu Mo**: Data curation. **Shu-Yu Luo**: Data curation. **Kun Liao**: Investigation. **Zhao-lei Zeng**: Investigation. **Shan-Shan Li**: Investigation. **Xin-Yuan Guan**: Investigation. **Xin-Juan Fan**: Investigation. **Hai-long Piao**: Formal analysis; Supervision; Methodology; Writing—review and editing. **Rui-Hua Xu**: Conceptualization; Supervision; Writing—review and editing. **Huai-Qiang Ju**: Conceptualization; Formal analysis; Supervision; Methodology; Writing—review and editing.

Source data underlying figure panels in this paper may have individual authorship assigned. Where available, figure panel/source data authorship is listed in the following database record: biostudies:S-SCDT-10_1038-S44321-024-00169-0.

## Disclosure and competing interests statement

The authors declare no competing interests. The authors have applied for patents for the use of the serum lipid metabolic signature to diagnose and predict biosamples.

# Expanded View Figures

**Figure EV1.  The analysis of preliminary experiment data and quality control data.**

(**A**) PCA on the lipid data or hydrophilic metabolite data of GC patients ($n = 28$) and healthy donors ($n = 28$). (**B**) PCA on the participant samples and QC samples showed that the QC samples were highly correlated. (**C**) Spearman's correlation coefficients between QC runs, ranging from 0.97 to 1, demonstrated the high stability and reproducibility of data. (**D**) Intensity distribution of lipid species indicated that QC samples ($n = 49$) had good consistency with participant samples in quantification of serum lipid levels; The sample numbers of GC, HD, and PL groups have been shown in Fig. 1A. (**E**) The validity of the partial least squares-discriminant analysis in Fig. 1B showed no overfitting (permutation test, $n = 1000$). Q2 measures the predictive ability of the model, while R2Y measures the goodness of fit. (**F**) The significantly changed lipids between GC patients and healthy donors. The classes of lipids are displayed in different colors. The black circle indicates 0 of the lipid level and the height of the bar represents the normalized lipid levels. The direction of bars pointing towards and away from the center represents the lipid level of healthy donors and GC patients, respectively. (**G**) The pathway enriched by the significantly changed lipid in serums (Hypergeometric test). The definitions of box plots in (**A**) and (**D**) were consistent with those in Fig. 3A,B. PCA principal component analysis, PC principal component, QC quality control, GC gastric cancer, HD healthy donor, PL precancerous lesion.

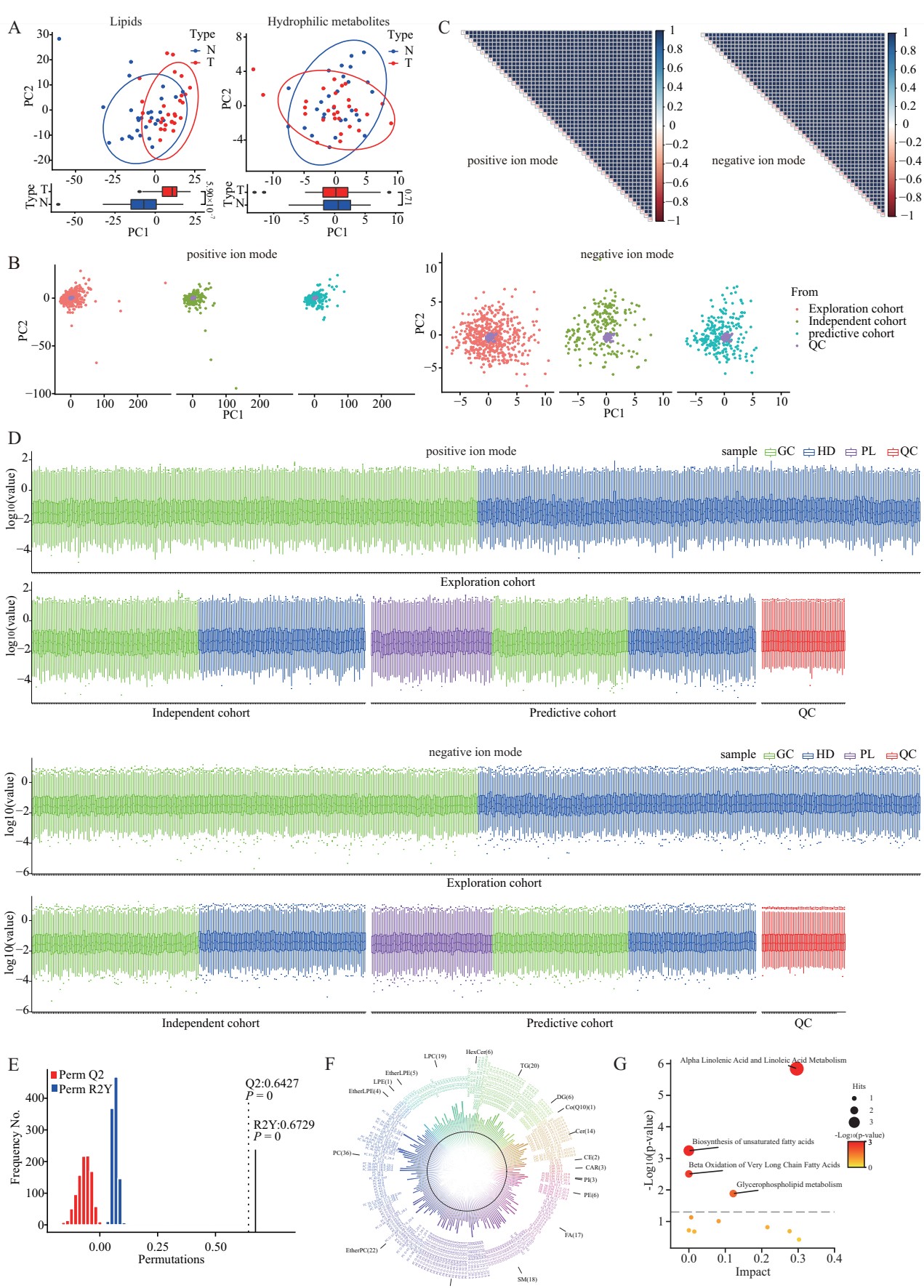

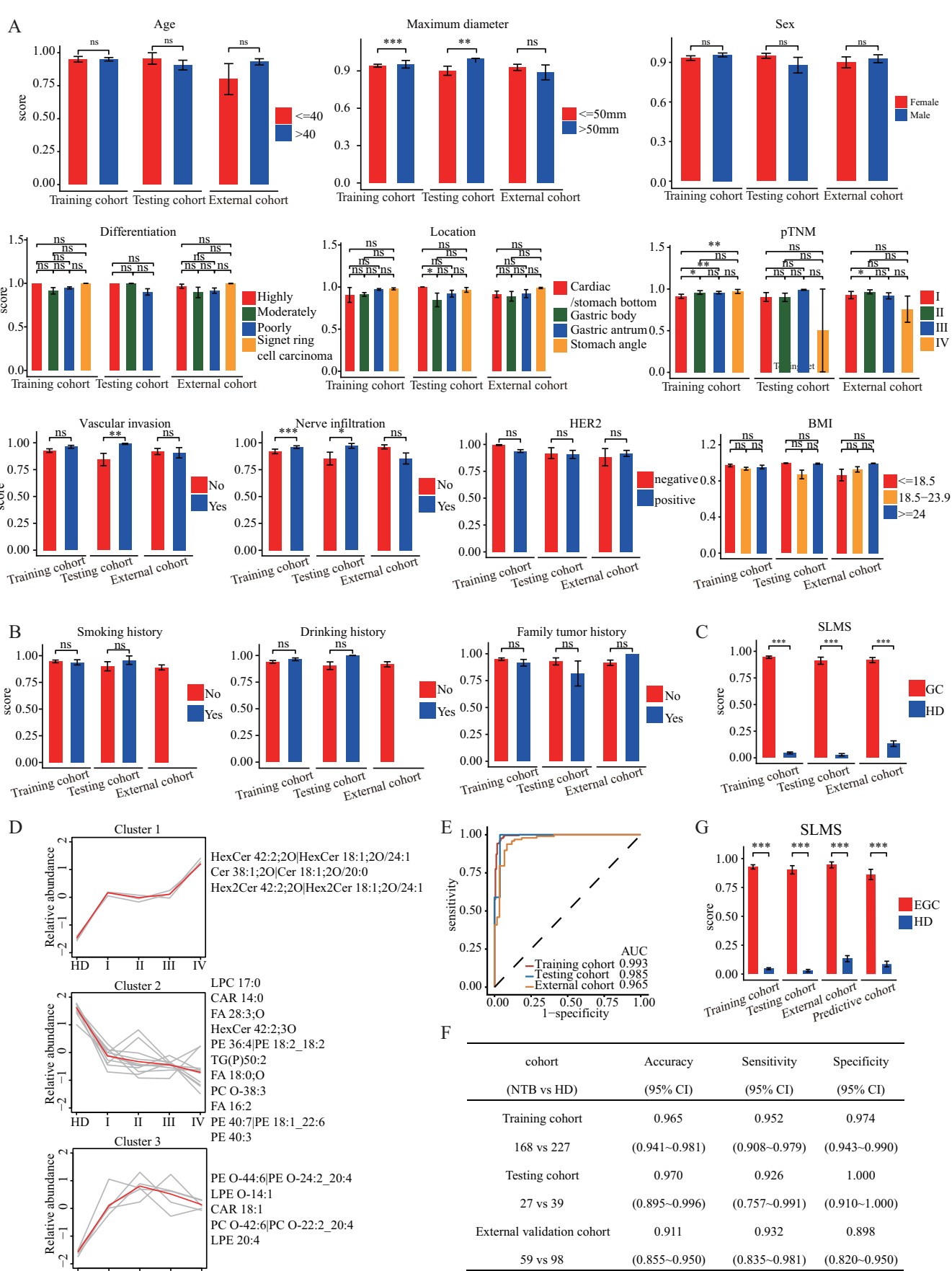

◀ **Figure EV2. The influence factor of the score of SLMS.**

(A) The SLMS scores of GC patients were compared between different stratification of age, maximum diameter, sex, differentiation, location, pTNM, vascular invasion, nerve infiltration, HER2 and BMI in the training, testing and external validation cohorts. (B) The SLMS scores of GC patients were compared between different stratification of smoking history, drinking history and family tumor history in the training, testing, and external validation cohorts. (C) The difference between the SLMS score of GC patients and that of HDs in the training, testing, and external validation cohorts. (D) Mfuzz clustering of lipid trajectories during GC progression using 19 lipids according to the lipid changes' similarity. Lipids in each cluster are presented on the side. HD, healthy donor. (E, F) The diagnostic performance of SLMS when used in detecting GC patients with negative CEA, CA19-9, and CA72-4. (G) The difference between the SLMS score of EGC patients and that of HDs in the training, testing, external validation, and predictive cohorts. $P$ values were determined by Wilcox test and Data presented as the mean ± S.D. (A, B, C, G). CA19-9 carbohydrate antigen 199, CA72-4 carbohydrate antigen 724, CEA carcinoembryonic antigen, CI confidence interval, EGC early-stage gastric cancer, GC gastric cancer, HD healthy donor, NTB negative for three biomarkers, SLMS serum lipid metabolic signature, ns non-significant; ***$P < 0.001$; **$P < 0.01$; *$P < 0.05$. Source data are available online for this figure.

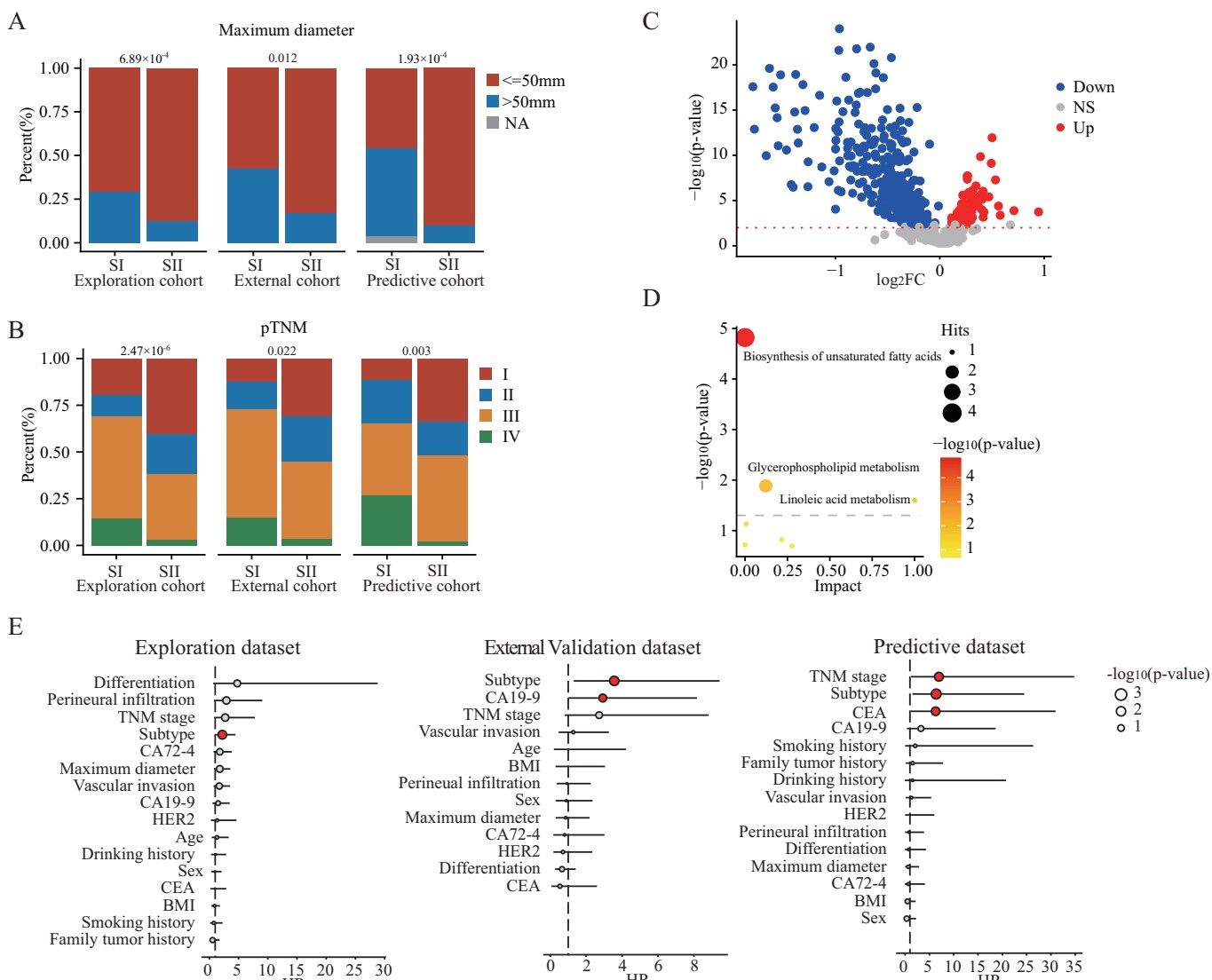

**Figure EV3. Characterization of GCPS.**

(A) The overlap between GCPS and maximum diameter. (B) The overlap between GCPS and pTNM stage. (C) Volcano plot comparing different lipids between SI and SII. (D) Enrichment analysis of different lipids between SI and SII. Hits are indicated by the size of the circle and significance is indicated by the color of the circle. (E) Multivariate Cox proportional hazards analyses of OS in patients with gastric cancer in three cohorts. The circles in red color indicated the P value was less than 0.05. P values were determined by Chi-square test (A, B), Wilcox test (C), Hypergeometric test (D), and Wald test (E). GCPS gastric cancer prognostic subtype, NS non-significant, OS overall survival, pTNM pathological Tumor-Node-Metastasis.

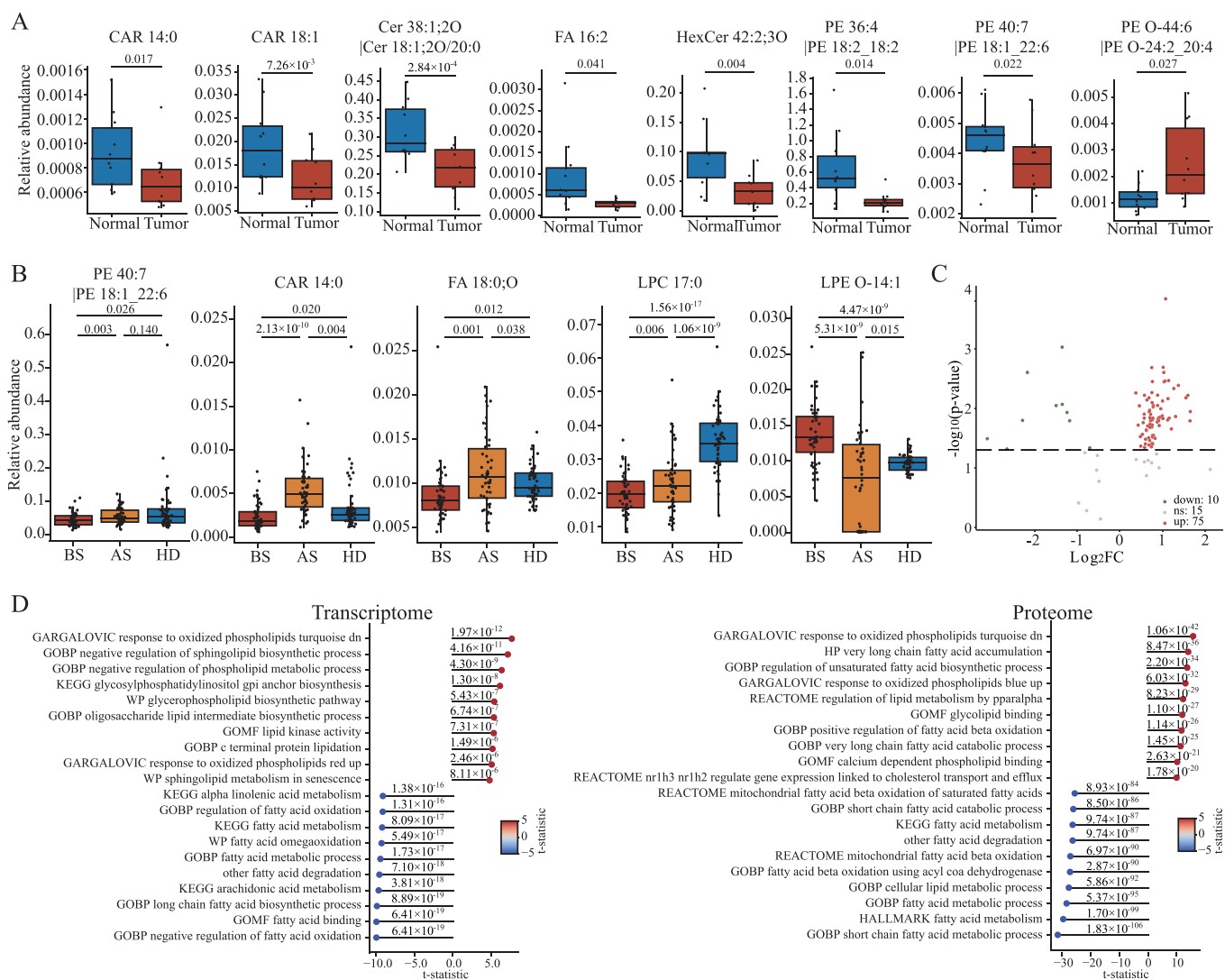

## Figure EV4. Analysis of the metabolites in SLMS.

(A) The significantly changed metabolites between gastric cancer ($n = 10$) and normal tissues ($n = 10$). (B) Metabolites that partially or completely return to normal levels after surgery ($n = 50$ per group). (C) Volcano plot showing the lipids that were differentially expressed between GC and normal regions. (D) Top 10 lipid-related metabolic pathways highly expressed in cancer tissues and normal tissues according to the transcriptome and proteome analysis. *P* values were determined by T test (A–D) and adjusted via the Benjamini-Hochberg procedure (D). The definitions of box plots in (A) and (B) were consistent with those in Fig. 3A,B. AS after surgery, BS before surgery, HD healthy donor, SLMS serum lipid metabolic signature.

No.1 patient with gastric cancer

No.2 patient with gastric cancer

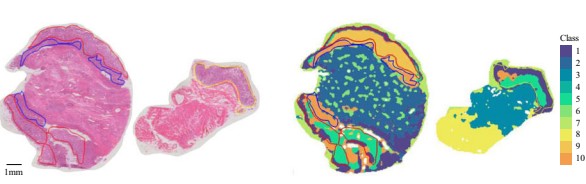

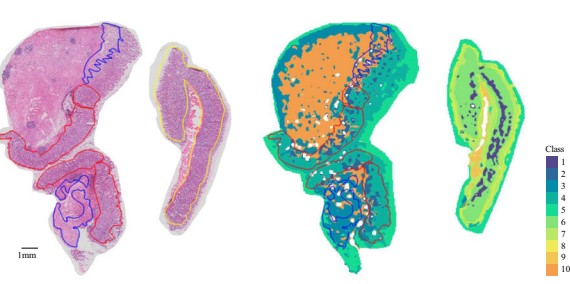

No.4 patient with gastric cancer

No.5 patient with gastric cancer

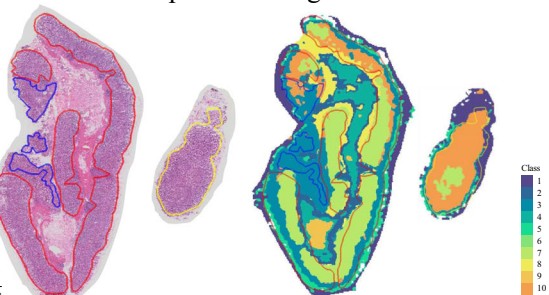

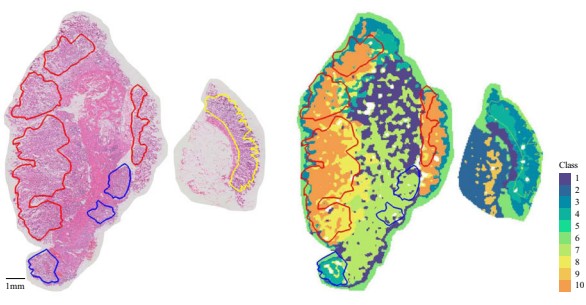

No.6 patient with gastric cancer

No.7 patient with gastric cancer

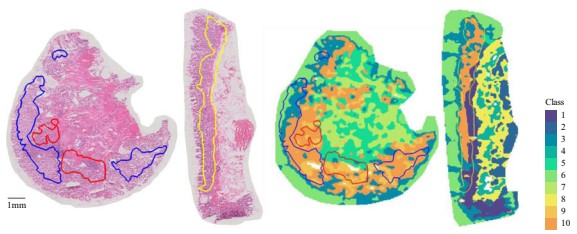

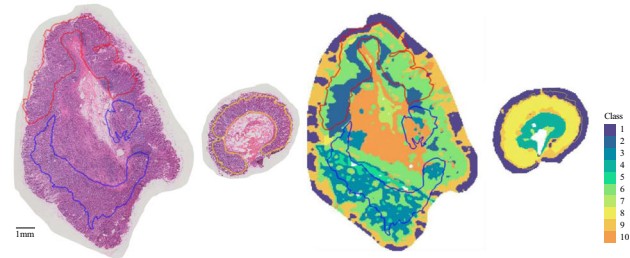

No.8 patient with gastric cancer

No.9 patient with gastric cancer

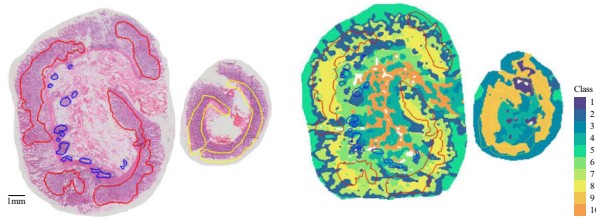

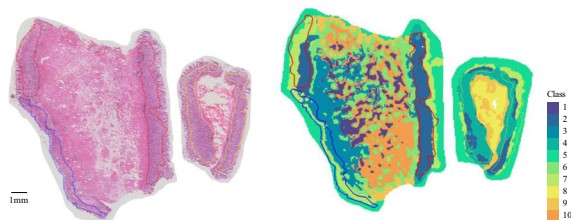

No.10 patient with gastric cancer

No.11 patient with gastric cancer

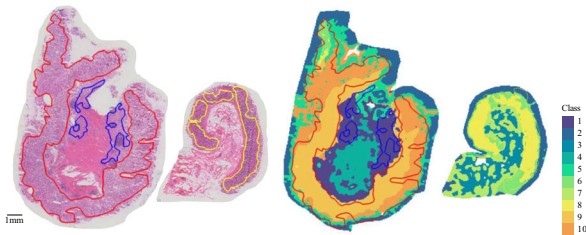

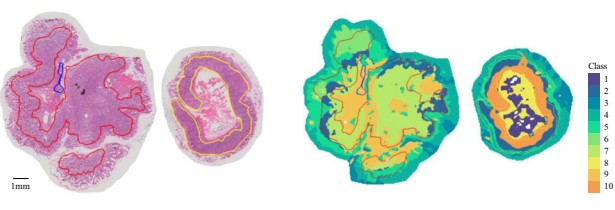

◀ **Figure EV5.** **The global metabolic landscape of patients with gastric cancer.**

H&E stain image and metabolite-driven segmentation of contiguous gastric cancer tissue sections. Scale bar = 1 mm. The blue, red, and yellow areas represent tumor, paratumor, and normal regions, respectively.

