## [Peer Review File · EMBO Molecular Medicine]

Diagnosis and Prognosis Prediction of Gastric Cancer by High-Performance Serum Lipidome Fingerprints

Ze-Rong Cai, Wen Wang, Di Chen, Hao-Jie Chen, Yan Hu, Yi-Ting Wang, Yi-Qian Pan, Hai-yu Mo, Shu-Yu Luo, Xiao-Jing Luo, Kun Liao, Zhao-lei Zeng, Shan-Shan Li, Xin-Yuan Guan, Xin-Juan Fan, Hai-long Piao, Rui-Hua Xu, and Huai-Qiang Ju

Corresponding authors: Huai-Qiang Ju (juhq@sysucc.org.cn), Rui-Hua Xu (xurh@sysucc.org.cn), Hai-long Piao (hpiao@dicp.ac.cn)

Review Timeline:

Submission Date:	2nd Aug 24
Editorial Decision:	29th Aug 24
Revision Received:	25th Sep 24
Editorial Decision:	11th Oct 24
Revision Received:	23rd Oct 24
Accepted:	25th Oct 24

Editor: Jingyi Hou

Transaction Report:

29th Aug 2024

Dear Dr. Ju,

Thank you again for submitting your work to EMBO Molecular Medicine. We have now heard back from the three referees who agreed to evaluate your manuscript. As you will see from the reports below, the referees find your study of potential interest. However, they raise a series of concerns, which should be convincingly addressed in a major revision of the present manuscript.

The referees' recommendations are relatively straightforward, so there is no need for me to reiterate the points listed below. All the issues raised by the reviewers need to be carefully addressed.

We would welcome the submission of a revised version within three months for further consideration. As you may already know, our editorial policy allows in principle a single round of major revision, and it is therefore essential to provide responses to the reviewers' comments that are as complete as possible.

Please also contact us as soon as possible if similar work is published elsewhere. If other work is published, we may not be able to extend the revision period beyond three months.

I look forward to receiving your revised manuscript.

Kind regards,
Jingyi

Jingyi Hou
Editor
EMBO Molecular Medicine

We require:

2) Individual production quality figure files as .eps, .tif, .jpg (one file per figure). For guidance, download the 'Figure Guide PDF': (<https://www.embopress.org/page/journal/17574684/authorguide#figureformat>).

3) A .docx formatted letter INCLUDING the reviewers' reports and your detailed point-by-point responses to their comments. As part of the EMBO Press transparent editorial process, the point-by-point response is part of the Review Process File (RPF), which will be published alongside your paper.

4) A complete author checklist, which you can download from our author guidelines (<https://www.embopress.org/page/journal/17574684/authorguide#submissionofrevisions>). Please insert information in the checklist that is also reflected in the manuscript. The completed author checklist will also be part of the RPF.

6) It is mandatory to include a 'Data Availability' section after the Materials and Methods. Before submitting your revision, primary datasets produced in this study need to be deposited in an appropriate public database, and the accession numbers and database listed under 'Data Availability'. Please remember to provide a reviewer password if the datasets are not yet public (see <https://www.embopress.org/page/journal/17574684/authorguide#dataavailability>).

12) Author contributions: You will be asked to provide CRediT (Contributor Role Taxonomy) terms in the submission system. These replace a narrative author contribution section in the manuscript.

13) A Conflict of Interest statement should be provided in the main text.

14) Every published paper now includes a 'Synopsis' to further enhance discoverability. Synopses are displayed on the journal webpage and are freely accessible to all readers. They include a short stand first (maximum of 300 characters, including space) as well as 2-5 one-sentences bullet points that summarizes the paper. Please write the bullet points to summarize the key NEW

findings. They should be designed to be complementary to the abstract - i.e. not repeat the same text. We encourage inclusion of key acronyms and quantitative information (maximum of 30 words / bullet point). Please use the passive voice. Please attach these in a separate file or send them by email, we will incorporate them accordingly.

15) All Materials and Methods need to be described in the main text using our 'Structured Methods' format, which is required for all research articles. According to this format, the Methods section includes a Reagents and Tools Table (listing key reagents, experimental models, software and relevant equipment and including their sources and relevant identifiers) followed by a Methods and Protocols section describing the methods using a step-by-step protocol format.

The Reagents and Tools Table can be downloaded from our author guidelines (<https://www.embopress.org/page/journal/17574684/authorguide#structuredmethods>)

***** Reviewer's comments *****

Referee #1 (Comments on Novelty/Model System for Author):

See remarks below.

Referee #1 (Remarks for Author):

In this work, Cai et al. present a comprehensive study on the use of serum lipid metabolic signatures for the diagnosis and prognosis prediction of gastric cancer. They integrate lipidomics data from multiple cohorts to develop a serum lipid metabolic signature (SLMS) for gastric cancer diagnosis and a gastric cancer prognostic subtype (GCPS) for prognosis prediction. The authors demonstrate that these lipid-based tools outperform traditional biomarkers in detecting gastric cancer, especially early-stage disease, and in stratifying patient outcomes. The authors should discuss specific strategies for integrating SLMS and GCPS into existing clinical decision-making processes, potentially proposing a stratified screening scheme that demonstrates how these new tools complement current diagnostic methods.

1. The study would benefit from a more thorough discussion of the potential confounding factors that could influence serum lipid profiles, such as BMI, diet, and medication use. The authors should explain how these factors were controlled for in the analysis.
2. While the authors concluded that scores showed no significant difference across various clinical characteristics, Figure EV2A suggests some potential variations, particularly for age and sex in the external cohort. A more detailed analysis of these potential variations could provide insights into the generalizability of the SLMS.
3. The authors reported that the top 50 lipids showed high correlations and only lipids with correlation coefficients less than 0.5 were used in later analysis. However, it is unclear whether the filtered top 50 lipids or a subset of these were used for further machine learning analysis. The rationale for using only lipids with correlation coefficients less than 0.5 in later analysis should be explained in more detail.
4. To provide a more comprehensive and fair comparison with clinically used biomarkers like CEA, CA19-9, and CA72-4, the authors should consider evaluating the performance of these biomarkers in combination.
5. The authors should consider including a more detailed comparison of the SLMS performance across different gastric cancer stages. This could provide valuable insights into the tool's utility across different stages. Also, more related literatures (View, 2023, 4, 20220038) should be included and discussed.
6. The authors should conduct a power analysis to ensure that the sample size used for model training is sufficient.
7. It would be valuable to include a more detailed description of any parameter tuning processes for the machine learning algorithms used.
8. The authors should consider discussing the potential for overfitting in their machine learning models and describe any measures taken to mitigate this risk.

Referee #2 (Comments on Novelty/Model System for Author):

This manuscript described a methodology for early diagnosis of gastric cancer and highlighted a novel lipid panel as the potential biomarkers. The biomarkers might be useful in the clinical diagnosis. The mass spectrometry-based metabolomics has been employed in this study to display the technical robustness.

Referee #2 (Remarks for Author):

This study introduces innovative liquid biopsy tools based on serum lipid metabolic fingerprints for diagnosis and prognosis of gastric cancer. The researchers have demonstrated a commendable dedication to their work by collecting serum samples from a considerable cohort of gastric cancer patients and healthy donors. The application of UPLC/MS technology has allowed them to identify crucial lipids in the serum that are significant for diagnosis and prognosis of gastric cancer. The development of the serum lipid metabolic signature is a noteworthy achievement, as it has shown promising results in clinical validation, surpassing the performance of traditional tumor markers in the early detection of gastric cancer. Besides, the gastric cancer prognostic subtype is desirable as it could indicate the prognosis of patients before surgery and provide guidance of postoperative treatment. Overall, this advancement not only offers hope for earlier screening but also highlights the potential of liquid biopsy in cancer diagnosis and prognosis assessment. Nevertheless, the manuscript has a few concerns to be addressed before publication.

Comments:

1. The authors utilized untargeted lipidomics at the beginning of the study and employed targeted lipidomics subsequently. Why don't they start with the targeted lipidomics or use the untargeted lipidomics throughout?
2. The authors mentioned "Lipid features were obtained with mass-to-charge ratio (m/z), retention time and MS/MS pattern by searching acquired MS/MS spectra against the internal MS/MS LipidBlast library in MS-DIAL program" in the part of data processing. However, they did not properly discuss how the identification of lipids was performed. It is recommended to disclose the mass accuracy, specific MS/MS fragments used to assign lipid class and fatty acyl rests.
3. The description of model building and selection process is insufficient. Ten commonly used classification have been utilized in lipidomics analysis. Is there thorough tuning of all algorithms? If yes, it would be better to mention the hyperparameters used for tuning.
4. The method used for the detection of serum samples after surgery was not clearly described. It is suggested to make a supplementary instruction about it.
5. Some issues about the spatial metabolome should be discussed, including how to identify or annotate MSI features, how to confirm these annotations with MS/MS, which features identified through MSI were utilized for clustering and the conclusion drawn from the clustering analysis of MSI images.
6. Some abbreviations were not written out in full. This practice should be rectified to improve the readability of the manuscript.

Referee #3 (Comments on Novelty/Model System for Author):

In the manuscript, Cai et al presented compelling data to propose lipidomics approaches for the early diagnosis and prognostic evaluation of gastric cancer. By using machine learning, the authors constructed the lipid signature that demonstrated outstanding performance (AUC>0.95) for detecting gastric cancer in multiple cohorts. Besides, the signature also exhibited excellent performance in diagnosing patients with early-stage GC. Alongside the diagnostic aspect, the authors also investigated serum lipid metabolites as potential indicators of GC prognosis. They uncovered changes in lipid metabolism in GC and observed variations in metabolic patterns linked to different GC prognosis subtypes. Furthermore, the authors conducted multi-omics to clarify the lipid disturbance in GC. Overall, this work is innovative and promising for clinical application.

Referee #3 (Remarks for Author):

There are several concerns which should be addressed before publication.

1. Although the scores of signature were independent of pTNM stages, it would be valuable to analyze the change or the trend of the lipids among different pTNM stages.
2. As mentioned in the article, CEA, CA19-9 and CA72-4 are not suitable for the screening of gastric cancer owing to their low sensitivities. Furthermore, it has been reported that 20-40% of gastric cancer patients are negative for all these three biomarkers due to the Lewis a-b- genotype, hence resulting in missed diagnoses. It is still unclear whether this new diagnostic method properly detect these patients.
3. Many studies have reported novel biomarkers from body fluids for the diagnosis of gastric cancer, such as ctDNA. Maybe it would be more interesting to contrast these already reported biomarkers with the SLMS constructed by the authors.
4. The authors did not indicate the median survival of different subtypes in Figure 4C, which may be caused by the good prognosis of patients with subtype I. Therefore, other indexes, such as two-year overall survival, are recommended to elucidate the different prognosis of different subtypes.
5. The SLMS contained 19 lipids, but the authors only showed the expression of 8 lipids in gastric cancer tissues in Figure EV4A. The remaining 11 lipids have not yet been discussed.
6. It is suggested to clarify the inclusion criteria of serum samples after surgery in Figure EV4B.
7. It would be beneficial to proof the manuscript carefully and modify the grammar errors. "As supplementary" should be corrected to "As a supplementary". "transform our results to clinical practice" should be corrected to "transform our results into clinical practice".

Manuscript ID: EMM-2024-20384

Title: Diagnosis and Prognosis Prediction of Gastric Cancer by High-Performance Serum Lipidome Fingerprints

Author Responses to Initial Comments:

Detailed point-by-point responses to the reviewers' comments:

Note to reviewers: We would like to express our sincere gratitude to the reviewers for their valuable time and constructive feedback on our manuscript. Their insightful comments have significantly contributed to enhancing the quality and scientific rigor of our work. In response to the reviewers' feedback, we have diligently revised the manuscript and incorporated new data to address all their concerns. Below are our point-by-point responses to each of the reviewers' comments. All changes in our revised manuscript have been marked in colored text. Besides, as suggested by the editors, we included the share link (<https://ngdc.cnbc.ac.cn/omix/preview/dIZvRZgt>) for reviewers to check the lipidomics data.

Referee #1 (Comments on Novelty/Model System for Author):

See remarks below.

Referee #1 (Remarks for Author):

In this work, Cai et al. present a comprehensive study on the use of serum lipid metabolic signatures for the diagnosis and prognosis prediction of gastric cancer. They integrate lipidomics data from multiple cohorts to develop a serum lipid metabolic signature (SLMS) for gastric cancer diagnosis and a gastric cancer prognostic subtype (GCPS) for prognosis prediction. The authors demonstrate that these lipid-based tools outperform traditional biomarkers in detecting gastric cancer, especially early-stage disease, and in stratifying patient outcomes. The authors should discuss specific strategies for integrating SLMS and GCPS into existing clinical

decision-making processes, potentially proposing a stratified screening scheme that demonstrates how these new tools complement current diagnostic methods.

Responses: We thank this reviewer for the thoughtful review of our manuscript, the positive and insightful comments, and constructive suggestions below to strengthen our hypothesis. As we all know, clinically effective management of cancers demands the integration of early detection with risk stratified interventions, and our study showed that serum lipid fingerprints could be applied in both early diagnosis and risk stratification. The SLMS could be used in recognizing potential GC patients, who would be confirmed by endoscopic biopsy finally. This strategy could improve the accuracy and efficiency of early screening. Subsequently, GC patients were classified into two groups with high and low risk of poor survival through the GCPS. The postoperative therapeutic regimen of them will be formulated with overall consideration of pTNM stages and GCPS. We have included this discussion in the revised manuscript (**Page 13 line 326-333, Page 14 line 359-366**). We hope that this interpretation satisfies the reviewer's query and clarifies the clinical application strategies of our new tools.

1. The study would benefit from a more thorough discussion of the potential confounding factors that could influence serum lipid profiles, such as BMI, diet, and medication use. The authors should explain how these factors were controlled for in the analysis.

Responses: We thank the reviewer for this outstanding question and valuable suggestions. Indeed, it is possible that these confounding factors could interfere with the serum lipid profile. Therefore, we analyzed the impact of BMI on the SLMS. The result showed that the scores of SLMS showed no significant difference across different stratifications of BMI in the training, testing and external validation cohorts (**Figure EV2A**). Besides, all participants had their blood drawn after fasting for at least eight hours, which mitigated the short-term effects of diet on the serum lipid profile. Nevertheless, the long-term shaping of serum lipid profiles by dietary habits has not been considered in this study. We have acknowledged this limitation in the

“Discussion” subsection. Furthermore, the serum samples were collected when the participants had not received any medication therapy. The relevant descriptions have been supplemented in the revised manuscript (**Page 15 line 380-381; Page 16 line 403-404, 406-409**). We hope that the reviewer will approve of our explanation.

Figure EV2A in the original manuscript

2. While the authors concluded that scores showed no significant difference across various clinical characteristics, Figure EV2A suggests some potential variations, particularly for age and sex in the external cohort. A more detailed analysis of these potential variations could provide insights into the generalizability of the SLMS.

Responses: We thank the reviewer for this outstanding question and valuable suggestions. To address this issue, we analyzed the distribution of age and sex in three cohorts. As shown in **the following two tables** and **Source data for Figure 1**, the age and sex distributions were balanced in the training and testing cohorts owing to the rigorous sample selection, but they were unbalanced in the validation cohort. Nevertheless, the SLMS performed well in both balanced and unbalanced cohorts.

Moreover, the SLMS was independent of age and sex in the training cohort and testing cohort owing to the balanced distribution (**Figure below A**). As pointed out by the reviewer, in the external validation cohort, the scores of SLMS were significantly lower in patients under 40 years old or female patients than that in patients over 40 years old or male patients (**Figure below A**). This is because there was a higher proportion of healthy donors in participants under the age of 40 or females (**the following two tables**), and the SLMS scores of healthy donors were

significantly lower than those of GC patients (**Figure EV2C**). Thus, we analyzed the relationship between SLMS scores and age or sex only in GC patients or only in healthy donors. As shown in **Figure below B**, the scores showed no significant difference across different stratifications of age or sex in GC patients, highlighting that the SLMS was not affected by age or sex. A similar result was found in healthy donors (**Figure below C**). Although the SLMS scores were significantly different between the two groups of healthy donors, the difference did not affect the judgment of sample types as the scores of two groups were both lower than 0.5 and samples with scores lower than 0.5 would be recognized as health. In general, the SLMS was independent of age and sex when used for judging the sample types. To avoid possible confusion, we have modified the content of Figure EV2 and included relevant description in the revised manuscript (**Page 6 line 148-149**). We hope that the reviewer will concur with our interpretation.

Table. The age distribution of participants in three cohorts.

	Age	GC patients	healthy donors	P value ^a
Training cohort	<=40	44	43	>0.999
	>40	183	184	
Testing cohort	<=40	4	9	0.224
	>40	35	30	
Validation cohort	<=40	10	76	<0.001
	>40	88	22	

^a*P* values were calculated by Fisher's exact test.

Table. The sex distribution of participants in three cohorts.

	Sex	GC patients	healthy donors	P value ^a
Training cohort	Female	114	111	0.851
	Male	113	116	
Testing cohort	Female	18	23	0.365
	Male	21	16	
Validation cohort	Female	36	50	0.061
	Male	62	48	

^a*P* values were calculated by Fisher's exact test.

Figure legend:

(A) The SLMS scores of all participants were compared between different stratification of age or sex in three cohorts.

(B) The SLMS scores of GC patients were compared between different stratification of age or sex in three cohorts.

(C) The SLMS scores of healthy donors were compared between different stratification of age or sex in three cohorts.

Figure EV2C in the original manuscript

3. The authors reported that the top 50 lipids showed high correlations and only lipids with correlation coefficients less than 0.5 were used in later analysis. However, it is unclear whether the filtered top 50 lipids or a subset of these were used for further machine learning analysis. The rationale for using only lipids with correlation coefficients less than 0.5 in later analysis should be explained in more detail.

Responses: We thank the reviewer for pointing out this issue and we sincerely apologize for our unclear description of method. Partial least squares-discriminant analysis (PLS-DA) were performed based on the scaled and pre-processed metabolism data of the training cohort and the lipids were ranked based on the variable importance projection (VIP) scores. The spearman correlation coefficients (SCCs) between the expressions of two lipids were calculated. The result revealed a high correlation between detected lipids, especially the top 50 lipids with the highest VIP scores (**Figure 1C-D**). To avoid duplicated information for training machine learning models, we proposed a step-wise feature selection strategy. Initially, the top one ranked lipid was selected. Then, the next lipid along the rank was checked sequentially and included only if the SCCs were less than 0.5 between this lipid and all the already included lipids; otherwise, this lipid was removed. Model training was performed based on the filtered top 50 lipids and different numbers of lipids from 2 to 50 were tried. We have included the detailed method in the revised manuscript (**Page 18 line 455-467**).

Besides, elimination of redundant features is a common method of feature selection in the construction of machine learning models (Ibrahim *et al*, 2021; Yu & Liu, 2003). If the redundant features are not removed, the performance of the model

will be degraded (**Figure below A, accuracy < 0.93 for top 19 lipids and LDA algorithm**) in comparison to the result reported in the original manuscript (**Figure 1E, accuracy = 0.963 for 19 lipids and LDA algorithm**). Meanwhile, we have previously compared the results with thresholds of 0.5, 0.6 and 0.7. The results showed that the accuracies were higher with the threshold of 0.5 than that of 0.6 or 0.7 (**Figure below B**). Therefore, we decided to filter the lipids using the threshold of 0.5. We hope this explanation will resolve the reviewer's confusion.

Figure 1C-E in the original manuscript

Figure legend:

(A) The accuracies of different models based on the top-n features (n from 2 to 50) without removing the lipids with high SCCs.

(B) Boxplot of the accuracies of different models based on the features, which were filtered through different thresholds of spearman correlation coefficients

4. To provide a more comprehensive and fair comparison with clinically used biomarkers like CEA, CA19-9, and CA72-4, the authors should consider evaluating the performance of these biomarkers in combination.

Responses: We appreciate the valuable suggestion raised by the reviewer. In view of the low sensitivities of CEA, CA19-9 and CA72-4(Sekiguchi & Matsuda, 2020), we proposed a combined diagnostic panel (CDP) based on these three markers, which could improve the sensitivities of clinically used biomarkers. Specifically, participants

who were positive for any of these three biomarkers would be identified as patients with gastric cancer. Then, we evaluated the performance of the CDP. The results showed that the sensitivities of the CDP were 0.260, 0.308 and 0.398 in the training, testing and external validation cohorts (Table below), and the efficacy of our SLMS still overpowered that of CDP greatly. We have inserted these results in Table 1 and the revised manuscript (**Page 7 line 173-177; Page 7 line 185-187**).

Table. The performance of the combined diagnostic panel based on CEA, CA19-9 and CA72-4.

cohort	Accuracy(95%CI)	Sensitivity(95%CI)	Specificity(95%CI)
Training cohort	0.549 (0.501~0.595)	0.260 (0.204~0.322)	0.837 (0.782~0.883)
Testing cohort	0.564 (0.447~0.676)	0.308 (0.170~0.476)	0.821 (0.665~0.925)
External validation cohort	0.622 (0.551~0.691)	0.398 (0.300~0.502)	0.847 (0.760~0.912)

In addition, it has been reported that 20-40% of gastric cancer patients are negative for all these three biomarkers due to the Lewis a⁻b⁻ genotype, hence resulting in missed diagnoses(Guo *et al*, 2023). To investigate the diagnostic efficacy of the SLMS for gastric cancer patients with negative CEA, CA19-9 and CA72-4, we screened out this population of GC patients from each cohort and assessed the performance of SLMS when used for judging them. As shown in **new Figure EV2E-F**, the SLMS revealed excellent diagnostic performance in the training, testing and external validation cohorts. These results suggested that the SLMS was superior to clinically used biomarkers and have been added in the revised manuscript (**Page 8 line 196-203**).

F

cohort (NTB vs HD)	Accuracy (95% CI)	Sensitivity (95% CI)	Specificity (95% CI)
Training cohort 168 vs 227	0.965 (0.941–0.981)	0.952 (0.908–0.979)	0.974 (0.943–0.990)
Testing cohort 27 vs 39	0.970 (0.895–0.996)	0.926 (0.757–0.991)	1.000 (0.910–1.000)
External validation cohort 59 vs 98	0.911 (0.855–0.950)	0.932 (0.835–0.981)	0.898 (0.820–0.950)

Figure legend:

(EV 2E-F) The diagnostic performance of SLMS when used in detecting GC patients with negative CEA, CA19-9 and CA72-4. NTB, negative for three biomarkers; CI, confidence interval.

5. The authors should consider including a more detailed comparison of the SLMS performance across different gastric cancer stages. This could provide valuable insights into the tool's utility across different stages. Also, more related literatures (View, 2023, 4, 20220038) should be included and discussed.

Responses: We thank the reviewer for this valuable suggestion. As suggested, we analyzed the efficacy of SLMS in diagnosing GC patients with different pTNM stages. The results showed that the SLMS showed similar high-performance in distinguishing patients with different pTNM stages from healthy donors (new Table EV5). It should be noted that in the testing and external validation cohorts, the efficiency of SLMS in diagnosing GC patients in stage IV was not as good as that in other stages owing to the number of patients in stage IV was small.

Table EV5. The performance of SLMS in distinguishing GC patients with different pTNM stages from healthy donors.

Cohort	Stage (N)	AUC	Accuracy (95% CI)	Sensitivity (95% CI)	Specificity (95% CI)
Training cohort	I (73)	0.990	0.963 (0.935~0.982)	0.932 (0.847~0.977)	0.974 (0.943~0.990)
	II (36)	0.998	0.973 (0.946~0.989)	0.972 (0.855~0.999)	0.974 (0.943~0.990)
	III (100)	0.994	0.973 (0.948~0.987)	0.970 (0.915~0.994)	0.974 (0.943~0.990)
	IV (18)	0.999	0.975 (0.948~0.991)	1.000 (0.815~1.000)	0.974 (0.943~0.990)
Testing cohort	I (13)	1.000	0.980 (0.897~1.000)	0.923 (0.640~0.998)	1.000 (0.910~1.000)
	II (11)	1.000	1.000 (0.929~1.000)	1.000 (0.715~1.000)	1.000 (0.910~1.000)
	III (13)	1.000	1.000 (0.932~1.000)	1.000 (0.753~1.000)	1.000 (0.910~1.000)
	IV (2)	0.795	0.976 (0.871~0.999)	0.500 (0.013~0.987)	1.000 (0.910~1.000)
External validation cohort	I (23)	0.962	0.909 (0.843~0.954)	0.957 (0.781~0.999)	0.898 (0.820~0.950)
	II (20)	0.989	0.915 (0.850~0.959)	1.000 (0.832~1.000)	0.898 (0.820~0.950)
	III (47)	0.969	0.903 (0.843~0.946)	0.915 (0.796~0.976)	0.898 (0.820~0.950)
	IV (8)	0.892	0.887 (0.811~0.940)	0.750 (0.349~0.968)	0.898 (0.820~0.950)
Predictive cohort	I (20)	0.944	0.921 (0.845~0.968)	0.850 (0.621~0.968)	0.942 (0.858~0.984)
	II (15)	0.987	0.941 (0.867~0.980)	0.933 (0.681~0.998)	0.942 (0.858~0.984)
	III (33)	0.988	0.951 (0.889~0.989)	0.970 (0.842~0.999)	0.942 (0.858~0.984)
	IV (8)	1.000	0.948 (0.872~0.986)	1.000 (0.631~1.000)	0.942 (0.858~0.984)

Abbreviations: SLMS, serum lipid metabolic signature; GC, gastric cancer; AUC, area under curve; CI, confidence interval.

In addition, the literature mentioned by the reviewer provided a systematic and comprehensive summary on the topic of machine learning-assisted mass spectrometry or spectroscopy for in vitro diagnosis(Chen *et al*, 2023), and our study was a new

representative example in this field. This literature has demonstrated that the development of technology and assistance of machine learning would advance the application of in vitro diagnosis, which has provided a solid theoretical foundation for our study. We have included and discussed this literature in the revised manuscript (**Page 4 line 81-84**).

6. The authors should conduct a power analysis to ensure that the sample size used for model training is sufficient.

Responses: We thank the reviewer for providing this valuable suggestion. We have used G*power 3.1.9.7 for a priori estimation of the sample size(Li *et al*, 2023). The result showed that the number of one group should exceed 64 when using the following parameters (two-tailed; effect size, 0.5; α , 0.05; power, 0.8). The sample sizes for training the diagnostic model and the prognostic subtype were 227 per group and 266 per group, respectively. In conclusion, the sample size in our study was sufficient. We have included the interpretation in the revised manuscript (**Page 16 line 402-403**).

7. It would be valuable to include a more detailed description of any parameter tuning processes for the machine learning algorithms used.

Responses: We thank the reviewer for pointing out this issue. Parameters of each algorithm were optimized by crossover validation and grid-search strategies. Concretely, the hyper-parameters of each algorithm, if any, were optimized by the train function in the caret package with default settings, this function can fit predictive models over different tuning parameters. The best tuning parameters for the algorithms were listed in **new Source Data 2 for Figure 1**. The relevant description has been included in the Materials and Methods (**Page 18 line 473-478**).

Source Data 2 for Figure 1. The hyper-parameters for all algorithms.

Algorithm	Best feature number	Final parameters	Tuned parameters	Average accuracy of 10-fold crossover
-----------	---------------------	------------------	------------------	---------------------------------------

				validation
RF	19	mtry=2	mtry=2	0.9512
			mtry=10	0.9175
			mtry=19	0.9094
SVMRadial	23	C=1, sigma=0.02878378	C=0.25	0.9441
			C=0.5	0.9551
			C=1	0.9578
SVMLinear	18	C=1	C=1	0.9516
SVMRadial	23	C=1,	C=0.25, Weight=1	0.9271
Weights		sigma=0.02666481, Weight=1	C=0.25, Weight=2	0.9250
			C=0.25, Weight=3	0.9159
			C=0.5, Weight=1	0.9537
			C=0.5, Weight=2	0.9427
			C=0.5, Weight=3	0.9335
			C=1, Weight=1	0.9566
			C=1, Weight=2	0.9498
			C=1, Weight=3	0.9452
KNN	10	k=9	k=5	0.9209
			k=7	0.9264
			k=9	0.9304
LDA	19	NO	NO	0.9635
Glmnet	24	alpha=0.55, lambda = 0.007395554	alpha=0.10, lambda =0.0007395554	0.9480
			alpha=0.10, lambda =0.0073955541	0.9502
			alpha=0.10, lambda =0.0739555410	0.9423
			alpha=0.55, lambda =0.0007395554	0.9482
			alpha=0.55, lambda =0.0073955541	0.9557
			alpha=0.55, lambda =0.0739555410	0.9469
			alpha=1.00, lambda =0.0007395554	0.9471
			alpha=1.00, lambda =0.0073955541	0.9526
			alpha=1.00, lambda =0.0739555410	0.9084
SVMLinear	18	cost=0.25,	cost=0.25, weight=1	0.9483
Weights		weight=1	cost=0.25, weight=2	0.9449
			cost=0.25, weight=3	0.9397
			cost=0.5, weight=1	0.9468
			cost=0.5, weight=2	0.9454
			cost=0.5, weight=3	0.9379
			cost=1, weight=1	0.9459
			cost=1, weight=2	0.9461
			cost=1, weight=3	0.9375
Bayesglm	20	NO	NO	0.9552
QDA	12	NO	NO	0.9499

Abbreviations: RF, random forest; SVMRadial, support vector machine with radial basis function; SVMLinear, linear support vector machine; SVMRadialWeights,

support vector machine with radial basis function and class weights; KNN, k-nearest neighbor; LDA, linear discrimination analysis; Glmnet, lasso and elastic-net regularized generalized linear model; SVMLinearWeights, linear support vector machine with class weights; Bayesglm, Bayesian generalized linear model; QDA, quadratic discriminant analysis.

8. The authors should consider discussing the potential for overfitting in their machine learning models and describe any measures taken to mitigate this risk.

Responses: We thank this reviewer for raising this crucial question. The diagnostic model was constructed based on the lipidomics data of the training cohort in a 10-fold cross-over validation manner. As shown in **Figure 1F**, the model achieved the performance with an average accuracy of 0.963 when the LDA algorithm was applied on the filtered top 19 metabolites. Besides, the diagnostic model showed high efficacy in differentiating between GC patients and healthy donors with AUCs of 0.989, 0.965 and 0.977 in the testing, external validation and predictive cohorts (**Figure 2B-C** and **Figure 3C**). Simultaneously, the model, constructed for predicting the prognosis of GC patients and based on the lipidomics dataset in the exploration cohort, successfully classified the patients into two groups with significantly different survival in the external validation cohort and the predictive cohort (**Figure 4C**). All these results indicated that there was no overfitting in our diagnostic and prognostic models. We have included the relevant description in the revised manuscript (**Page 7 line 184-185; Page 10 line 265-267**).

Figure 1F in the original manuscript

	Accuracy	Specificity	Sensitivity	AUC
mean	0.963	0.958	0.971	0.988
95%CI	0.958–0.969	0.951–0.965	0.964–0.978	0.985–0.991

Figure 2C and 2D in the original manuscript

Figure 3C in the original manuscript

Figure 4C in the original manuscript

Referee #2 (Comments on Novelty/Model System for Author):

This manuscript described a methodology for early diagnosis of gastric cancer and highlighted a novel lipid panel as the potential biomarkers. The biomarkers might be

useful in the clinical diagnosis. The mass spectrometry-based metabolomics has been employed in this study to display the technical robustness.

Responses: We thank this reviewer for the thoughtful review of our manuscript.

Referee #2 (Remarks for Author):

This study introduces innovative liquid biopsy tools based on serum lipid metabolic fingerprints for diagnosis and prognosis of gastric cancer. The researchers have demonstrated a commendable dedication to their work by collecting serum samples from a considerable cohort of gastric cancer patients and healthy donors. The application of UPLC/MS technology has allowed them to identify crucial lipids in the serum that are significant for diagnosis and prognosis of gastric cancer. The development of the serum lipid metabolic signature is a noteworthy achievement, as it has shown promising results in clinical validation, surpassing the performance of traditional tumor markers in the early detection of gastric cancer. Besides, the gastric cancer prognostic subtype is desirable as it could indicate the prognosis of patients before surgery and provide guidance of postoperative treatment. Overall, this advancement not only offers hope for earlier screening but also highlights the potential of liquid biopsy in cancer diagnosis and prognosis assessment. Nevertheless, the manuscript has a few concerns to be addressed before publication.

Responses: We appreciate for the positive and insightful comments, and constructive suggestions from this reviewer. And we hope that our revision has largely addressed the concerns from this reviewer.

Comments:

1. The authors utilized untargeted lipidomics at the beginning of the study and employed targeted lipidomics subsequently. Why don't they start with the targeted lipidomics or use the untargeted lipidomics throughout?

Responses: This reviewer raised an outstanding question. In fact, more lipids could be detected by untargeted lipidomics, which ensures the comprehensiveness of lipid

detection. The quantification of lipids by targeted lipidomics is more accurate. The validation of SLMS by targeted lipidomics further verified the efficiency of SLMS. Therefore, we started our study with untargeted lipidomics following by targeted lipidomics. We hope that this interpretation satisfies the reviewer's query and clarifies our experimental design.

2. The authors mentioned "Lipid features were obtained with mass-to-charge ratio (m/z), retention time and MS/MS pattern by searching acquired MS/MS spectra against the internal MS/MS LipidBlast library in MS-DIAL program" in the part of data processing. However, they did not properly discuss how the identification of lipids was performed. It is recommended to disclose the mass accuracy, specific MS/MS fragments used to assign lipid class and fatty acyl rests.

Response: We thank the reviewer for pointing out this issue. In this study, **accurate mass tolerance (MS1)** and **accurate mass tolerance (MS2)** were 0.01Da and 0.025Da, respectively. For lipids without ionic fragments but appeared in the full scan mass spectrogram, the identity of these lipid candidates was further confirmed by comparing the relative retention time between the known lipids and the candidate peaks within the same lipid class. **For specific MS/MS fragments used to assign lipid class**, for example, 184.0739 was selected as a characteristic product ion for PC, LPC, PC-O, LPC-O, and SM. 369.3516 was selected as a characteristic product ion for CE. 266.2791, 264.2635, 262.2479 and 312.326 were selected as characteristic product ions of Cer or HexCer with skeletons d18:0, d18:1, d18:2, and d20:0, respectively. 241.0119 was selected as a characteristic product ion for PI. **For acyl chains**, 251.2016, 283.2642, 281.2485, 279.2329, 303.2325, 301.2169 and 327.2329 were taken as the characteristics of appearance of fatty acyls 16:2, 18:0, 18:1, 18:2, 20:4, 20:5 and 22:6, respectively. We have provided clearer descriptions of how to identify the lipids in the Extended View Methods.

3. The description of model building and selection process is insufficient. Ten commonly used classification have been utilized in lipidomics analysis. Is there thorough tuning of all algorithms? If yes, it would be better to mention the

hyperparameters used for tuning.

Responses: We thank the reviewer for this outstanding question. Parameters of each algorithm were optimized by crossover validation and grid-search strategies.

Concretely, the hyper-parameters of each algorithm, if any, were optimized by the train function in the caret package with default settings, this function can fit predictive models over different tuning parameters. The best tuning parameters for all algorithms were listed in **new Source Data 2 for Figure 1**. In addition, we have included more details about the optimization of the algorithms in the revised manuscript (**Page 18 line 473-478**).

Source Data 2 for Figure 1. The hyper-parameters for all algorithms.

Algorithm	Best feature number	Final parameters	Tuned parameters	Average accuracy of 10-fold crossover validation
RF	19	mtry=2	mtry=2	0.9512
			mtry=10	0.9175
			mtry=19	0.9094
SVMRadial	23	C=1, sigma=0.02878378	C=0.25	0.9441
			C=0.5	0.9551
			C=1	0.9578
SVMLinear	18	C=1	C=1	0.9516
SVMRadial Weights	23	C=1, sigma=0.02666481, Weight=1	C=0.25, Weight=1	0.9271
			C=0.25, Weight=2	0.9250
			C=0.25, Weight=3	0.9159
			C=0.5, Weight=1	0.9537
			C=0.5, Weight=2	0.9427
			C=0.5, Weight=3	0.9335
			C=1, Weight=1	0.9566
			C=1, Weight=2	0.9498
KNN	10	k=9	k=5	0.9209
			k=7	0.9264
			k=9	0.9304
LDA	19	NO	NO	0.9635
Glmnet	24	alpha=0.55, lambda = 0.007395554	alpha=0.10, lambda =0.0007395554	0.9480
			alpha=0.10, lambda =0.0073955541	0.9502
			alpha=0.10, lambda =0.0739555410	0.9423

			alpha=0.55, lambda =0.0007395554	0.9482
			alpha=0.55, lambda =0.0073955541	0.9557
			alpha=0.55, lambda =0.0739555410	0.9469
			alpha=1.00, lambda =0.0007395554	0.9471
			alpha=1.00, lambda =0.0073955541	0.9526
			alpha=1.00, lambda =0.0739555410	0.9084
SVMLinear	18	cost=0.25,	cost=0.25, weight=1	0.9483
Weights		weight=1	cost=0.25, weight=2	0.9449
			cost=0.25, weight=3	0.9397
			cost=0.5, weight=1	0.9468
			cost=0.5, weight=2	0.9454
			cost=0.5, weight=3	0.9379
			cost=1, weight=1	0.9459
			cost=1, weight=2	0.9461
			cost=1, weight=3	0.9375
Bayesglm	20	NO	NO	0.9552
QDA	12	NO	NO	0.9499

Abbreviations: RF, random forest; SVMRadial, support vector machine with radial basis function; SVMLinear, linear support vector machine; SVMRadialWeights, support vector machine with radial basis function and class weights; KNN, k-nearest neighbor; LDA, linear discrimination analysis; Glmnet, lasso and elastic-net regularized generalized linear model; SVMLinearWeights, linear support vector machine with class weights; Bayesglm, Bayesian generalized linear model; QDA, quadratic discriminant analysis.

4. The method used for the detection of serum samples after surgery was not clearly described. It is suggested to make a supplementary instruction about it.

Responses: We thank the reviewer for providing this valuable suggestion. We have collected the preoperative serum samples from 50 GC patients, paired postoperative serum samples from the same 50 patients and serum samples from 50 healthy donors. Besides, the postoperative serum samples were collected within three months after surgery and before any postoperative adjuvant therapy. We have examined the expression of SLMS lipids in these samples by utilizing untargeted lipidomics, which has been detailly described in the Materials and Methods. As suggested, we have included related description in the revised manuscript (**Page 18 line 450-451**).

5. Some issues about the spatial metabolome should be discussed, including how to identify or annotate MSI features, how to confirm these annotations with MS/MS, which features identified through MSI were utilized for clustering and the conclusion drawn from the clustering analysis of MSI images.

Responses: We thank the reviewer for pointing out these issues. It should be clarified that metabolite identification was performed by comparing the MS and MS/MS spectrum information with the in-house database and the Human Metabolome Database (HMDB). Besides, extracted adducted ions without MS/MS fragmentation information were imported into in-house database and the HMDB and annotated for lipid species according to molecular weight with an error $< \pm 10$ ppm. In addition, all confirmed target peaks were used to conduct spatially-aware nearest shrunken centroids clustering to obtain ten partition. As shown in **Figures 5B** and **Figure EV5**, each partition was represented by a color and different color-labeled partitions represented different metabolic patterns. Furthermore, adjacent tissue sections of samples undergoing mass spectrometry image (MSI) were stained with hematoxylin-eosin (**Figure 5A**). The region of gastric cancer, paratumor and normal were encircled by blue, red and yellow lines respectively. Through integrating hematoxylin-eosin images with the clustering analysis of MSI images, it could be found that there were different metabolic patterns among the tumor, paratumor, and normal regions. In response to the reviewer's concern, we have included these descriptions in Extended View Methods.

Figure 5A and 5B in the original manuscript

6. Some abbreviations were not written out in full. This practice should be rectified to

improve the readability of the manuscript.

Responses: We sincerely apologize for any confusion caused by the unclear description of abbreviations. As suggested, we have rearranged the abbreviations in the main text and attached them in a separate file .dox file named “Synopsis and Acronyms”

Referee #3 (Comments on Novelty/Model System for Author):

In the manuscript, Cai et al presented compelling data to propose lipidomics approaches for the early diagnosis and prognostic evaluation of gastric cancer. By using machine learning, the authors constructed the lipid signature that demonstrated outstanding performance ($AUC > 0.95$) for detecting gastric cancer in multiple cohorts. Besides, the signature also exhibited excellent performance in diagnosing patients with early-stage GC. Alongside the diagnostic aspect, the authors also investigated serum lipid metabolites as potential indicators of GC prognosis. They uncovered changes in lipid metabolism in GC and observed variations in metabolic patterns linked to different GC prognosis subtypes. Furthermore, the authors conducted multi-omics to clarify the lipid disturbance in GC. Overall, this work is innovative and promising for clinical application.

Responses: We thank the reviewer for appreciating our study and providing valuable comments to enhance the quality of our manuscript. We hope that our revision has largely addressed the critiques raised by this reviewer.

Referee #3 (Remarks for Author):

There are several concerns which should be addressed before publication.

1. Although the scores of signature were independent of pTNM stages, it would be valuable to analyze the change or the trend of the lipids among different pTNM stages.

Responses: We thank the reviewer for providing this valuable suggestion. As suggested, we have conducted the cluster analysis on the trend of 19 lipids in the SLMS. Interestingly, these 19 lipids showed 3 significantly distinct trends (Cluster 1-3) along with the progression of GC (**new Figure EV2D**). Specifically, the lipids in Cluster 1 (e.g., HexCer 42:2;2O|HexCer 18:1;2O/24:1) exhibited a continuously increasing pattern while those lipids in Cluster 2 (e.g., LPC 17:0) showed a sustainably decreasing trend along with cancer development. Besides, some lipids, such as PE O-44:6|PE O-24:2_20:4, partially declined after climbing up during the progression of GC. We have added a description in the revised manuscript (**Page 6 line 158-160; Page 7 line 161-165**).

Figure legend:

(EV 2D) Mfuzz clustering of lipid trajectories during GC progression using 19 lipids according to the lipid changes' similarity. Lipids in each cluster were presented on the side. HD, healthy donor.

2. As mentioned in the article, CEA, CA19-9 and CA72-4 are not suitable for the

screening of gastric cancer owing to their low sensitivities. Furthermore, it has been reported that 20-40% of gastric cancer patients are negative for all these three biomarkers due to the Lewis a-b- genotype, hence resulting in missed diagnoses. It is still unclear whether this new diagnostic method properly detect these patients.

Responses: We thank the reviewer for this outstanding question and valuable suggestions. To investigate the diagnostic efficacy of the SLMS for gastric cancer patients with negative CEA, CA19-9 and CA72-4, we screened out this population of gastric cancer patients from each cohort and assessed the performance of SLMS when used for judging them. As shown in **new Figure EV2E-F**, the SLMS revealed excellent diagnostic performance in the training, testing and external validation cohorts. These results have been added in the revised manuscript (**Page 8 line 196-203**).

F

cohort	Accuracy	Sensitivity	Specificity
(NTB vs HD)	(95% CI)	(95% CI)	(95% CI)
Training cohort	0.965	0.952	0.974
168 vs 227	(0.941~0.981)	(0.908~0.979)	(0.943~0.990)
Testing cohort	0.970	0.926	1.000
27 vs 39	(0.895~0.996)	(0.757~0.991)	(0.910~1.000)
External validation cohort	0.911	0.932	0.898
59 vs 98	(0.855~0.950)	(0.835~0.981)	(0.820~0.950)

Figure legend:

(EV 2E-F) The diagnostic performance of SLMS when used in detecting GC patients with negative CEA, CA19-9 and CA72-4. NTB, negative for three biomarkers; CI,

confidence interval.

3. Many studies have reported novel biomarkers from body fluids for the diagnosis of gastric cancer, such as ctDNA. Maybe it would be more interesting to contrast these already reported biomarkers with the SLMS constructed by the authors.

Responses: This reviewer asked a very insightful question. Indeed, it has been reported that ctDNAs and exosome-derived non-coding RNAs could act as novel diagnostic biomarkers for GC (Guo *et al.*, 2023; Maron *et al.*, 2019). Nevertheless, a small amount of ctDNAs in circulation require sensitive and expensive detection techniques while the fragmented exosome-derived ncRNAs could not reflect the overall genetic profile of the tumors. In contrast to DNA and RNA, lipids could directly reflect the holistic and real-time phenotypes of the tumors. Besides, the detection method of lipids, such as UPLC/MS, was universal and low-cost. Recent studies have showed that lipidomics could be utilized in the diagnosis of various cancers (Wang *et al.*, 2022; Wang *et al.*, 2021; Wolrab *et al.*, 2022). In parallel, the SLMS and the GCPS we constructed though lipidomics exhibited great performances in the diagnosis and prognostic prediction of GC. In conclusion, the innovative lipid fingerprints were valuable and promising for the diagnosis and prognosis of GC. This discussion has been elaborated upon in our revised manuscript (**Page 13 line 317-326**).

4. The authors did not indicate the median survival of different subtypes in Figure 4C, which may be caused by the good prognosis of patients with subtype I. Therefore, other indexes, such as two-year overall survival, are recommended to elucidate the different prognosis of different subtypes.

Responses: We thank the reviewer for pointing out this issue. As suggested, we have calculated two-year overall survival rates of two subtypes in each cohort. The result showed that in the exploration cohort, the two-year overall survival rates of patients with SI and SII were 75.7% and 88.8%, respectively (**new Table EV6**). The similar results could be found in the external validation cohort and the predictive cohort. Also,

we have included related description in the revised manuscript (**Page 10 line 250-253**).

Table EV6. The two-year overall survival of patients with different prognostic subtypes in each cohort.

	SI (two-year overall survival*)	SII (two-year overall survival*)
Exploration cohort	75.7 (67.6~84.7)	88.8(83.8~94.0)
External validation cohort	59.1 (45.5~76.7)	89.2(81.4~97.8)
Predictive cohort	58.6 (40.1~84.1)	91.4(83.7~99.9)

*Data are presented as % (95% confidence interval).

5. The SLMS contained 19 lipids, but the authors only showed the expression of 8 lipids in gastric cancer tissues in Figure EV4A. The remaining 11 lipids have not yet been discussed.

Responses: We are thankful to the reviewer for raising this question. Among the 19 lipids in SLMS, 8 lipids displayed remarkable differences in abundance between gastric cancer tissues and normal tissues, which have been shown in Figure EV4A. Nevertheless, the changes of other 8 lipids were not significant, which were shown in Table EV9.1. Additionally, FA 28:3;O, LPE O-14:1 and PE 40:3 could not be detected in tissues despite they could be measured in serums. We speculated that this is caused by inconsistent sample types. We hope that the reviewer will concur with our interpretation.

Table EV9.1 (in the original manuscript). Differential expression of metabolites in the SLMS between tumor and normal tissues from 11 GC patients.

metabolites	t statistic	P value ^a
LPC 17:0	-1.180	0.268
PE O-44:6	2.642	0.027
PE O-24:2_20:4		
CAR 14:0	-2.915	0.017
HexCer 42:2;2O	-1.401	0.195
HexCer 18:1;2O/24:1		

HexCer 42:2;3O	-3.810	0.004
PE 36:4 PE 18:2_18:2	-3.063	0.014
CAR 18:1	-3.452	0.007
Cer 38:1;2O	-5.729	<0.001
Cer 18:1;2O/20:0	-0.735	0.481
TG(P) 50:2	-0.739	0.479
FA 18:0;O	-0.132	0.898
PC O-38:3	-2.377	0.041
FA 16:2	2.047	0.071
PC O-42:6	1.839	0.099
PC O-22:2_20:4	-2.762	0.022
Hex2Cer 42:2;2O	-2.104	0.065
Hex2Cer 18:1;2O/24:1		
PE 40:7 PE 18:1_22:6		
LPE 20:4		

Abbreviations: GC, gastric cancer; SLMS, serum lipid metabolic signature.

aPaired t-test was used for comparing two groups.

6. It is suggested to clarify the inclusion criteria of serum samples after surgery in Figure EV4B.

Responses: We thank the reviewer for providing this valuable suggestion. The inclusion criteria of serum samples after surgery included (1) from the same patients as preoperative samples, (2) within three months after surgery, (3) before any postoperative adjuvant therapy. We have added a brief description in the revised manuscript (**Page 17 line 424-426**).

7. It would be beneficial to proof the manuscript carefully and modify the grammar errors. "As supplementary" should be corrected to "As a supplementary". "transform our results to clinical practice" should be corrected to "transform our results into clinical practice".

Responses: Thank you for pointing out this issue and we apologized for the mistakes. We have proofed our revised manuscript carefully and corrected the grammar errors.

References:

Chen X, Shu W, Zhao L, Wan J (2023) Advanced mass spectrometric and spectroscopic methods coupled with machine learning for in vitro diagnosis. *VIEW* 4: 20220038

Guo X, Peng Y, Song Q, Wei J, Wang X, Ru Y, Xu S, Cheng X, Li X, Wu D *et al* (2023) A Liquid Biopsy Signature for the Early Detection of Gastric Cancer in Patients. *Gastroenterology* 165: 402-413.e413

Ibrahim S, Nazir S, Velastin SA (2021) Feature Selection Using Correlation Analysis and Principal Component Analysis for Accurate Breast Cancer Diagnosis. *J Imaging* 7

Li K, Lin Y, Zhou Y, Xiong X, Wang L, Li J, Zhou F, Guo Y, Chen S, Chen Y *et al* (2023) Salivary Extracellular MicroRNAs for Early Detection and Prognostication of Esophageal Cancer: A Clinical Study. *Gastroenterology* 165: 932-945.e939

Maron SB, Chase LM, Lomnicki S, Kochanny S, Moore KL, Joshi SS, Landron S, Johnson J, Kiedrowski LA, Nagy RJ *et al* (2019) Circulating Tumor DNA Sequencing Analysis of Gastroesophageal Adenocarcinoma. *Clin Cancer Res* 25: 7098-7112

Sekiguchi M, Matsuda T (2020) Limited usefulness of serum carcinoembryonic antigen and carbohydrate antigen 19-9 levels for gastrointestinal and whole-body cancer screening. *Sci Rep* 10: 18202

Wang G, Qiu M, Xing X, Zhou J, Yao H, Li M, Yin R, Hou Y, Li Y, Pan S *et al* (2022) Lung cancer scRNA-seq and lipidomics reveal aberrant lipid metabolism for early-stage diagnosis. *Sci Transl Med* 14: eabk2756

Wang G, Yao H, Gong Y, Lu Z, Pang R, Li Y, Yuan Y, Song H, Liu J, Jin Y *et al* (2021)

Metabolic detection and systems analyses of pancreatic ductal adenocarcinoma

through machine learning, lipidomics, and multi-omics. *Sci Adv* 7: eabh2724

Wolrab D, Jirásko R, Cífková E, Höring M, Mei D, Chocholoušková M, Peterka O,

Idkowiak J, Hrnčiarová T, Kuchař L *et al* (2022) Lipidomic profiling of human serum

enables detection of pancreatic cancer. *Nat Commun* 13: 124

Yu L, Liu H (2003) *Feature Selection for High-Dimensional Data: A Fast*

Correlation-Based Filter Solution

11th Oct 2024

Dear Dr. Ju,

Thank you for submitting your revised manuscript to EMBO Molecular Medicine. We have now received the enclosed report from the three referees who re-assessed your work. As you will see, the referees are now supportive, and I am pleased to inform you that we will be able to accept your manuscript pending the following amendments:

Please address the remaining minor point raised by Referee #2.

On a more editorial level:

1. Please rename "Materials & Methods" to "Methods".
2. BIORENDER: Please remove the related information from Acknowledgments and add a "Graphics" section to the Methods following the format: Graphics: (some of the... OR Figure #... OR synopsis) Graphics were created with BioRender.com.
3. EV tables should be uploaded as separate files, with one file per table. Each file should include a separate sheet labeled 'Legend' containing the corresponding table legend.
4. Please merge the Appendix Supplementary Methods with the Methods section in the main manuscript file. Appendix should be removed.
5. Data citation: Please note that the data callouts in the text for "Shi W, et al" and "Cancer Genome Atlas Research Network (2014) Xenus" data citation does not include "Data ref:" as a prefix. Please see below for our data citation guideline:

Data citations in the article text are distinct from normal bibliographical citations and should directly link to the database records from which the data can be accessed. In the main text, data citations are formatted as follows: "Data ref: Smith et al, 2001". In the Reference list, data citations must be labeled with "[DATASET]". A data reference must provide the database name, accession number/identifiers and a resolvable link to the landing page from which the data can be accessed at the end of the reference. Further instructions are available at .

6. I have slightly modified 'The Paper Explained'(see attached). Please let me know if you are fine with it or if you would like to introduce further modifications.

7. Please address the following issues regarding figure legends:

- Please note that the figure 6d-e does not contain any p value, kindly rectify the statistical test related information in the figure legend appropriately.
- Please note that the figure legend is mislabeled as figure EV 4c as figure EV 4d for statistical test in the manuscript. This needs to be rectified.
- Please note that the exact p values are not provided in the legends of figures 3a-b; 4c-d; EV 1e; EV 2a, c, g; EV 3a-b; EV 4a-b, d.
- Please indicate the statistical test used for data analysis in the legends of figures EV 1d-e, g.
- Please note that in figures 3a-b, there is a mismatch between the annotated p values in the figure legend and the annotated p values in the figure file that should be corrected.
- Please note that the box plots need to be defined in terms of minima, maxima, centre, bounds of box and whiskers, and percentile in the legends of figures 3a-b; EV 1a, d; EV 4a-b.
- Please note that information related to n is missing in the legends of figures 3a-b; 5f; EV 1a, d; EV 2a-c, g.

Please submit your revised manuscript within two weeks. I look forward to seeing a revised form of your manuscript as soon as possible.

Kind regards,
Jingyi

Jingyi Hou
Editor
EMBO Molecular Medicine

*** Instructions to submit your revised manuscript ***

- 1) a .docx formatted version of the manuscript text (including Figure legends and tables)
 - 2) Separate figure files*
 - 3) supplemental information as Expanded View and/or Appendix. Please carefully check the authors guidelines for formatting Expanded view and Appendix figures and tables at <https://www.embopress.org/page/journal/17574684/authorguide#expandedview>
 - 4) a letter INCLUDING the reviewer's reports and your detailed responses to their comments (as Word file).
 - 5) The paper explained: EMBO Molecular Medicine articles are accompanied by a summary of the articles to emphasize the major findings in the paper and their medical implications for the non-specialist reader. Please provide a draft summary of your article highlighting
 - the medical issue you are addressing,
 - the results obtained and
 - their clinical impact.This may be edited to ensure that readers understand the significance and context of the research. Please refer to any of our published articles for an example.
 - 6) Author contributions: the contribution of every author must be detailed in a separate section.
 - 7) EMBO Molecular Medicine now requires a complete author checklist (<https://www.embopress.org/page/journal/17574684/authorguide>) to be submitted with all revised manuscripts. Please use the checklist as guideline for the sort of information we need WITHIN the manuscript. The checklist should only be filled with page numbers where the information can be found. This is particularly important for animal reporting, antibody dilutions (missing) and exact values and n that should be indicated instead of a range.
 - 8) Every published paper now includes a 'Synopsis' to further enhance discoverability. Synopses are displayed on the journal webpage and are freely accessible to all readers. They include a short stand first (maximum of 300 characters, including space) as well as 2-5 one sentence bullet points that summarise the paper. Please write the bullet points to summarise the key NEW findings. They should be designed to be complementary to the abstract - i.e. not repeat the same text. We encourage inclusion of key acronyms and quantitative information (maximum of 30 words / bullet point). Please use the passive voice. Please attach these in a separate file or send them by email, we will incorporate them accordingly.
- You are also welcome to suggest a striking image or visual abstract to illustrate your article. If you do please provide a jpeg file 550 px-wide x 300-600px high.
- 9) A Conflict of Interest statement should be provided in the main text
 - 10) Please note that we now mandate that all corresponding authors list an ORCID digital identifier. This takes <90 seconds to complete. We encourage all authors to supply an ORCID identifier, which will be linked to their name for unambiguous name identification.

Currently, our records indicate that the ORCID for your account is 0000-0003-1713-5465.

Link Not Available

11) Include a Reagents and Tools Table as part of the Methods section, which can be downloaded from our author guidelines (<https://www.embopress.org/page/journal/17574684/authorguide#structuredmethods>)

Photos 400-800 DPI

*Additional important information regarding figures and illustrations can be found at

<https://bit.ly/EMBOPressFigurePreparationGuideline>. See also figure legend preparation guidelines:

<https://www.embopress.org/page/journal/17574684/authorguide#figureformat>

***** Reviewer's comments *****

Referee #2 (Comments on Novelty/Model System for Author):

The combination of lipidomics and machine learning is very powerful in GC early diagnosis.

Referee #2 (Remarks for Author):

Line 72, "wildly" should be changed to "widely".

Referee #3 (Remarks for Author):

The authors have addressed all of my concerns-congratulations!

The authors addressed the remaining editorial issues.

25th Oct 2024

Dear Dr. Ju,

We are pleased to inform you that your manuscript is accepted for publication and is now being sent to our publisher to be included in the next available issue of EMBO Molecular Medicine.

Yours sincerely,
Jingyi

Jingyi Hou
Editor
EMBO Molecular Medicine
